# MULTI-ARMED BANDITS WITH ABSTENTION

## ABSTRACT

We introduce a novel extension of the canonical multi-armed bandit problem that incorporates an additional strategic element: *abstention*. In this enhanced framework, the agent is not only tasked with selecting an arm at each time step, but also has the option to abstain from accepting the stochastic instantaneous reward before observing it. When opting for abstention, the agent either suffers a fixed regret or gains a guaranteed reward. Given this added layer of complexity, we ask whether we can develop efficient algorithms that are both asymptotically and minimax optimal. We answer this question affirmatively by designing and analyzing algorithms whose regrets meet their corresponding information-theoretic lower bounds. Our results offer valuable quantitative insights into the benefits of the abstention option, laying the groundwork for further exploration in other online decision-making problems with such an option. Numerical results further corroborate our theoretical findings.

## 1 INTRODUCTION

In the realm of online decision-making, the multi-armed bandit model, originally introduced by Thompson (1933), has long served as a quintessential benchmark for capturing the delicate interplay between exploration and exploitation. In stochastic multi-armed bandit problems, the agent sequentially selects an arm from the given set at each time step and subsequently observes a random reward associated with the chosen arm. To maximize cumulative rewards, the agent must strike a balance between the persistent pursuit of the arm with the highest expected reward (exploitation) and the adventurous exploration of other arms to gain a deeper understanding of their potential (exploration). This fundamental challenge finds applications across a wide array of domains, ranging from optimizing advertising campaigns to fine-tuning recommendation systems.

However, real-world scenarios often come fraught with complexities that challenge the simplicity of the canonical bandit model. One notable complexity arises when the agent is equipped with an additional option to abstain from accepting the stochastic instantaneous reward before actually observing it. This added layer of decision-making considerably enriches the strategic landscape, altering how the agent optimally navigates the trade-off between exploration and exploitation.

Consider, for example, the domain of clinical trials. When evaluating potentially hazardous medical treatments, researchers can proactively deploy safeguards such as preemptive medications or consider purchasing specialized insurance packages to shield against possible negative consequences. However, these protective measures come with costs, which may be modeled as either fixed regrets or fixed rewards in the context of the clinical study's cumulative regret. In these scenarios, researchers have the option to observe the outcomes of a treatment while abstaining from incurring the associated random regret through these costly prearranged measures. Opting for abstention can promote more responsible decision-making and reduce the overall cumulative regret of the study.

Building upon this challenge, we introduce an innovative extension to the canonical multi-armed bandit model that incorporates abstention as a legitimate strategic option. At each time step, the agent not only selects which arm to pull but also decides whether to abstain. Depending on how the abstention option impacts the cumulative regret, which is the agent's primary optimization objective, our abstention model offers two complementary settings, namely, the fixed-regret setting where abstention results in a constant regret, and the fixed-reward setting where abstention yields a deterministic reward. Collectively, these settings provide the agent with a comprehensive toolkit for adeptly navigating the complicated landscape of online decision-making.

**Main contributions.** Our main results and contributions are summarized as follows:

  (i) In Section 2, we provide a rigorous mathematical formulation of the multi-armed bandit model with abstention. Our focus is on cumulative regret minimization across two distinct yet complementary settings: *fixed-regret* and *fixed-reward*. These settings give rise to divergent performance metrics, each offering unique analytical insights. Importantly, both settings encompass the canonical bandit model as a particular case.

 (ii) In the fixed-regret setting, we judiciously integrate two abstention criteria into a Thompson Sampling-based algorithm proposed by Jin et al. (2023). This integration ensures compatibility with the abstention option, as elaborated in Algorithm 1. The first abstention criterion employs a carefully constructed lower confidence bound, while the second is tailored to mitigate worst-case scenarios. We establish both asymptotic and minimax upper bounds on the cumulative regret. Furthermore, we derive corresponding lower bounds, thereby demonstrating that our algorithm attains asymptotic and minimax optimality simultaneously.

(iii) In the fixed-reward setting, we introduce a general strategy, outlined in Algorithm 2. This method is capable of transforming any algorithm that is both asymptotically and minimax optimal in the canonical model to one that also accommodates the abstention option. Remarkably, this strategy maintains its universal applicability and straightforward implementation while provably achieving both forms of optimality—asymptotic and minimax.

(iv) To empirically corroborate our theoretical contributions, we conduct a series of numerical experiments in Section 5. These experiments substantiate the effectiveness of our algorithms and highlight the performance gains achieved through the inclusion of the abstention option.

## 1.1 RELATED WORK

**Canonical multi-armed bandits.** The study of cumulative regret minimization in canonical multi-armed bandits has attracted considerable scholarly focus. Two dominant paradigms for evaluating optimality metrics emerge: asymptotic optimality and minimax optimality. Briefly, the former considers the behavior of algorithms as the time horizon approaches infinity for a specific problem instance, while the latter seeks to minimize the worst-case regret over all possible instances. A diverse array of policies have been rigorously established to achieve asymptotic optimality across various settings. Notable examples include UCB2 (Auer et al., 2002), DMED (Honda & Takemura, 2010), KL-UCB (Cappé et al., 2013), and Thompson Sampling (Agrawal & Goyal, 2012; Kaufmann et al., 2012). In the context of the worst-case regret, MOSS (Audibert & Bubeck, 2009) stands out as the pioneering method that has been verified to be minimax optimal. Remarkably, KL-UCB$^{++}$ (Ménard & Garivier, 2017) became the first algorithm proved to achieve both asymptotic and minimax optimality. Very recently, Jin et al. (2023) introduced Less-Exploring Thompson Sampling, an innovation that boosts computational efficiency compared to classical Thompson Sampling while concurrently achieving asymptotic and minimax optimality. For a comprehensive survey of bandit algorithms, we refer to Lattimore & Szepesvári (2020).

**Machine learning with abstention.** Starting with the seminal works of Chow (1957; 1970), the concept of learning with *abstention* (also referred to as *rejection*) has been extensively explored in various machine learning paradigms. These include, but are not limited to, classification (Herbei & Wegkamp, 2006; Bartlett & Wegkamp, 2008; Cortes et al., 2016), ranking (Cheng et al., 2010; Mao et al., 2023), and regression (Wiener & El-Yaniv, 2012; Zaoui et al., 2020; Kalai & Kanade, 2021).

Within this broad spectrum of research, our work is most directly related to those that explore the role of abstention in the context of *online learning*. To the best of our knowledge, Cortes et al. (2018) firstly incorporated the abstention option into the problem of online prediction with expert advice (Littlestone & Warmuth, 1994). In their model, at each time step, each expert has the option to either make a prediction based on the given input or abstain from doing so. When the agent follows the advice of an expert who chooses to abstain, the true label of the input remains undisclosed, and the learner incurs a known fixed loss. Subsequently, Neu & Zhivotovskiy (2020) introduced a different abstention model, which is more similar to ours. Here, the abstention option is only available to the agent. Crucially, the true label is always revealed to the agent *after* the decision has been made, regardless of whether the agent opts to abstain. Their findings suggest that equipping the agent with an abstention option can significantly improve the guarantees on the worst-case regret.

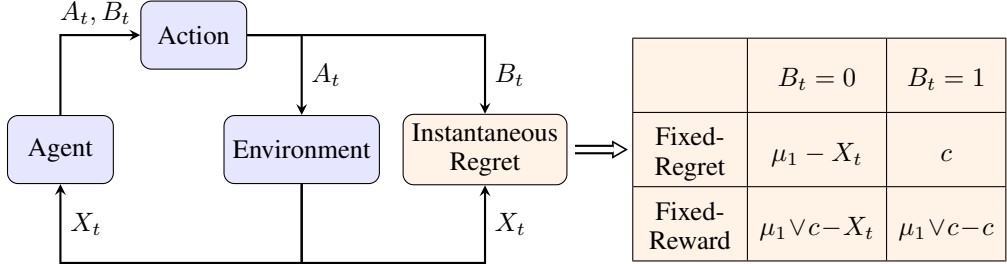

Figure 1: Interaction protocol for multi-armed bandits with fixed-regret and fixed-reward abstention.

Although set in different contexts, these existing works consistently demonstrate the value of incorporating abstention into online decision-making processes, underscoring the urgent need to analyze and quantify its benefits in the field of multi-armed bandits.

## 2 PROBLEM SETUP

**Multi-armed bandits with abstention.** We consider a $K$-armed bandit model, enhanced with an additional option to abstain from accepting the stochastic instantaneous reward prior to its observation. Let $\mu \in \mathcal{U} := \mathbb{R}^K$ denote a specific bandit instance, where $\mu_i$ represents the unknown mean reward associated with pulling arm $i \in [K]$. For simplicity, we assume that arm 1 is the unique optimal arm, i.e., $1 = \arg\max_{i \in [K]} \mu_i$, and we define $\Delta_i := \mu_1 - \mu_i$ as the suboptimality gap for each arm $i \in [K]$.

At each time step $t \in \mathbb{N}$, the agent chooses an arm $A_t$ from the given arm set $[K]$, and, simultaneously, decides whether or not to abstain, indicated by a binary variable $B_t$. Regardless of the decision to abstain, the agent observes a random variable $X_t$ from the selected arm $A_t$, which is drawn from a Gaussian distribution $\mathcal{N}(\mu_{A_t}, 1)$ and independent of observations obtained from the previous time steps. Notably, the selection of both $A_t$ and $B_t$ might depend on the previous decisions and observations, as well as on each other. More formally, let $\mathcal{F}_t := \sigma(A_1, B_1, X_1, \ldots, A_t, B_t, X_t)$ denote the $\sigma$-field generated by the cumulative interaction history up to and including time $t$. It follows that the pair of random variables $(A_t, B_t)$ is $\mathcal{F}_{t-1}$-measurable.

The instantaneous regret at time $t$ is determined by both the binary abstention variable $B_t$ and the observation $X_t$. Based on the outcome of the abstention option, we now discuss two complementary settings. In the *fixed-regret* setting, the abstention option incurs a constant regret. Opting for abstention ($B_t = 1$) leads to a deterministic regret of $c > 0$, in contrast to the initial regret linked to arm $A_t$ when not selecting abstention ($B_t = 0$), which is given by $\mu_1 - X_t$.

Alternatively, in the *fixed-reward* setting, the reward of the abstention option is predetermined to be $c \in \mathbb{R}$.[1] Since the abstention reward $c$ may potentially surpass $\mu_1$, the best possible expected reward at a single time step is $\mu_1 \vee c := \max\{\mu_1, c\}$. If the agent decides to abstain ($B_t = 1$), it guarantees a deterministic reward of $c$, leading to a regret of $\mu_1 \vee c - c$. Conversely, if $B_t = 0$, the agent receives a per-time reward $X_t$, resulting in a regret of $\mu_1 \vee c - X_t$.

See Figure 1 for a schematic of our model in the two settings.

**Regret minimization.** Our overarching goal is to design and analyze online algorithms $\pi$ that minimize their expected cumulative regrets up to and including the time horizon $T$.[2] The regrets are formally defined for the two distinct settings as follows:

- Fixed-regret setting:

$$R^{\text{RG}}_{\mu,c}(T, \pi) := \mathbb{E}\left[\sum_{t=1}^{T} \left((\mu_1 - X_t) \cdot \mathbb{1}\{B_t = 0\} + c \cdot \mathbb{1}\{B_t = 1\}\right)\right]. \quad (1)$$

---

[1]With a slight abuse of notation, we employ the symbol $c$ to represent the abstention regret and the abstention reward within their respective settings. The surrounding context, nonetheless, should elucidate the exact meaning of $c$.

[2]In certain real-world applications, the time horizon $T$ may be unknown to the agent. In fact, all of our proposed methods are inherently *anytime* in nature, as they do not necessitate prior knowledge of the horizon.

- Fixed-reward setting:

$$R_{\mu,c}^{\mathrm{RW}}(T,\pi) := T \cdot (\mu_1 \vee c) - \mathbb{E}\left[\sum_{t=1}^{T}\left(X_t \cdot \mathbb{1}\{B_t = 0\} + c \cdot \mathbb{1}\{B_t = 1\}\right)\right]. \quad (2)$$

An online algorithm $\pi$ consists of two interrelated components: the *arm sampling* rule that selects $A_t$, and the *abstention decision rule* that determines $B_t$ at each time step $t \in [T]$. Additionally, we use $\Pi^{\mathrm{RG}}$ and $\Pi^{\mathrm{RW}}$ to denote the collections of all online policies for the fixed-regret and fixed-reward settings, respectively. For the sake of analytical convenience, we also introduce the canonical regret $R_{\mu}^{\mathrm{CA}}(T,\pi) := T\mu_1 - \mathbb{E}\left[\sum_{t=1}^{T} X_t\right]$, which disregards the abstention option and remains well-defined within our abstention model. Furthermore, when there is no ambiguity, we will omit the dependence of the regret on the policy. For example, we often abbreviate $R_{\mu,c}^{\mathrm{RG}}(T,\pi)$ as $R_{\mu,c}^{\mathrm{RG}}(T)$.

*Remark* 1. It is worth mentioning that our model is a strict generalization of the canonical multi-armed bandit model (without the abstention option). Specifically, it particularizes to the canonical model as the abstention regret $c$ tends to positive infinity in the fixed-regret setting and as the abstention reward $c$ tends to negative infinity in the fixed-reward setting. Nevertheless, the incorporation of an extra challenge, the abstention decision (denoted as $B_t$), offers the agent the potential opportunity to achieve superior performance in terms of either regret.

**Other notations.** For $x, y \in \mathbb{R}$, we denote $x \wedge y := \min\{x,y\}$ and $x \vee y := \max\{x,y\}$. For any arm $i \in [K]$, let $N_i(t) := \sum_{s=1}^{t} \mathbb{1}\{A_s = i\}$ and $\hat{\mu}_i(t) := \sum_{s=1}^{t} X_s \mathbb{1}\{A_s = i\}/N_i(t)$ denote its total number of pulls and empirical estimate of the mean up to time $t$, respectively. In particular, we set $\hat{\mu}_i(t) = +\infty$ if $N_i(t) = 0$. To count abstention records, we also use $N_i^{(0)}(t)$ and $N_i^{(1)}(t)$ to denote its number of pulls without and with abstention up to time $t$, respectively. That is, $N_i^{(0)}(t) := \sum_{s=1}^{t} \mathbb{1}\{A_s = i \text{ and } B_s = 0\}$ and $N_i^{(1)}(t) := \sum_{s=1}^{t} \mathbb{1}\{A_s = i \text{ and } B_s = 1\}$. Additionally, we define $\hat{\mu}_{is}$ as the empirical mean of arm $i$ based on its first $s$ pulls. Furthermore, we use $\alpha, \alpha_1$, and so forth to represent universal constants that do not depend on the *problem instances* (including $\mu$, $c$, $T$, $K$), with possibly different values in different contexts.

## 3 FIXED-REGRET SETTING

In this section, we focus on the fixed-regret setting. Specifically, we design a conceptually simple and computationally efficient algorithm, namely Fixed-Regret Thompson Sampling with Abstention (or FRG-TSwA), to minimize the cumulative regret while incorporating fixed-regret abstention. To evaluate the performance of our algorithm from a theoretical standpoint, we establish both instance-dependent asymptotic and instance-independent minimax upper bounds on the cumulative regret, as elaborated upon in Section 3.1. Furthermore, in Section 3.2, we provide lower bounds for the problem of regret minimization in multi-armed bandits with fixed-regret abstention. These findings substantiate that our algorithm achieves both asymptotic and minimax optimality simultaneously. The pseudocode for FRG-TSwA is presented in Algorithm 1 and elucidated in the following.

In terms of the arm sampling rule, our algorithm is built upon Less-Exploring Thompson Sampling (Jin et al., 2023), a minimax optimal enhancement of the celebrated Thompson Sampling algorithm (Thompson, 1933). We refer to Remark 3 for the reason behind this choice. During the initialization phase, each arm is sampled exactly once. Following that, at each time $t$, an estimated reward $a_i(t)$ is constructed for each arm $i \in [K]$, which is either drawn from the posterior distribution $\mathcal{N}(\hat{\mu}_i(t-1), 1/N_i(t-1))$ with probability $1/K$ or set to be the empirical mean $\hat{\mu}_i(t-1)$ otherwise. Subsequently, the algorithm consistently pulls the arm $A_t$ with the highest estimated reward.

With regard to the abstention decision rule, we propose two abstention criteria that work in tandem (as detailed in Step 5 of Algorithm 1). The first criterion is gap-dependent in nature. In particular, we choose to abstain if there exists an arm $i \in [K] \setminus \{A_t\}$ for which the difference between its *lower confidence bound* and the empirical mean of the arm $A_t$ exceeds $c$. This condition signifies that the suboptimality gap $\Delta_{A_t}$ is at least $c$ with high probability. The second abstention criterion is gap-independent and more straightforward. It is motivated from the construction of worst-case scenarios as detailed in the proof of our lower bound. Under this criterion, we opt for the abstention option if $c \leq \sqrt{K/t}$, which implies that the abstention regret remains acceptably low at time $t$ in view of the worst-case scenarios.

---

**Algorithm 1** Fixed-Regret Thompson Sampling with Abstention (or FRG-TSwA)

**Input:** Arm set $[K]$ and abstention regret $c > 0$.

1: Sample each arm once, and choose to abstain ($B_t = 1$) if and only if $\sqrt{\frac{K}{t}} \geq c$.
2: Initialize $\hat{\mu}_i(K)$ and $N_i(K) = 1$ for all $i \in [K]$.
3: **for** $t = K + 1, \ldots, T$ **do**
4:     For each arm $i \in [K]$, sample $\theta_i(t) \sim \mathcal{N}(\hat{\mu}_i(t-1), 1/N_i(t-1))$ and set

$$a_i(t) = \begin{cases} \theta_i(t) & \text{with probability } 1/K \\ \hat{\mu}_i(t-1) & \text{with probability } 1 - 1/K. \end{cases}$$

5:     Pull the arm $A_t = \arg\max_{i \in [K]} a_i(t)$, and choose to abstain ($B_t = 1$) if and only if

$$\max_{i \in [K] \setminus \{A_t\}} \left( \hat{\mu}_i(t-1) - \sqrt{\frac{6 \log t + 2 \log(c \vee 1)}{N_i(t-1)}} \right) - \hat{\mu}_{A_t}(t-1) \geq c \text{ or } \sqrt{\frac{K}{t}} \geq c.$$

6:     Observe $X_t$ from the arm $A_t$, and update $\hat{\mu}_i(t)$ and $N_i(t)$ for all $i \in [K]$.
7: **end for**

---

## 3.1 Upper Bounds

Theorem 1 below provides two distinct types of theoretical guarantees pertaining to our algorithm's performance on the cumulative regret $R_{\mu,c}^{\text{RG}}(T)$, which is defined in Equation (1) for the fixed-regret setting. The complete proof of Theorem 1 is deferred to Appendix C.1.

**Theorem 1.** *For all abstention regrets $c > 0$ and bandit instances $\mu \in \mathcal{U}$, Algorithm 1 guarantees that*

$$\limsup_{T \to \infty} \frac{R_{\mu,c}^{\text{RG}}(T)}{\log T} \leq 2 \sum_{i>1} \frac{\Delta_i \wedge c}{\Delta_i^2}.$$

*Furthermore, there exists a universal constant $\alpha > 0$ such that*

$$R_{\mu,c}^{\text{RG}}(T) \leq \begin{cases} cT & \text{if } c \leq \sqrt{K/T} \\ \alpha(\sqrt{KT} + \sum_{i>1} \Delta_i) & \text{if } c > \sqrt{K/T}. \end{cases}$$

*Remark* 2. The theoretical challenges associated with Theorem 1 revolve around quantifying the regret that results from inaccurately estimating the suboptimality gaps associated to the abstention criteria. More precisely, from both asymptotic and worst-case perspectives, it is crucial to establish upper bounds on $\mathbb{E}[N_i^{(1)}(T)]$ for arms $i$ with $\Delta_i < c$ (which, by definition, includes the best arm), and on $\mathbb{E}[N_i^{(0)}(T)]$ for arms $i$ with $\Delta_i > c$. These complexities necessitate a deeper exploration into the arm sampling dynamics inherent to Less-Exploring Thompson Sampling, and preclude us from formulating a generalized strategy akin to the upcoming Algorithm 2 for the fixed-reward setting.

*Remark* 3. As previously highlighted, our model in the fixed-regret setting particularizes to the canonical multi-armed bandit model as the abstention regret $c$ approaches infinity. Similarly, when $c$ tends towards infinity, the two abstention criteria are never satisfied, and the procedure of Algorithm 1 simplifies to that of Less-Exploring Thompson Sampling. It is worth noting that this latter algorithm is not only asymptotically optimal but also minimax optimal for the canonical model. This is precisely why we base our algorithm upon it, rather than the conventional Thompson Sampling algorithm, which has been shown not to be minimax optimal (Agrawal & Goyal, 2017).

## 3.2 Lower Bounds

In order to establish the asymptotic lower bound, we need to introduce the concept of $R^{\text{RG}}$-consistency, which rules out overly specialized algorithms that are tailored exclusively to specific problem instances. Roughly speaking, a $R^{\text{RG}}$-consistent algorithm guarantees a subpolynomial cumulative regret for any given problem instance.

**Definition 1** ($R^{\text{RG}}$-consistency). An algorithm $\pi \in \Pi^{\text{RG}}$ is said to be $R^{\text{RG}}$-*consistent* if for all abstention regrets $c > 0$, bandit instances $\mu \in \mathcal{U}$, and $a > 0$, $R_{\mu,c}^{\text{RG}}(T, \pi) = o(T^a)$.

Now we present both asymptotic and minimax lower bounds on the cumulative regret in Theorem 2, which is proved in Appendix C.2.

**Theorem 2.** *For any abstention regret $c > 0$, bandit instance $\mu \in \mathcal{U}$ and $R^{\mathrm{RG}}$-consistent algorithm $\pi$, it holds that*

$$\liminf_{T \to \infty} \frac{R^{\mathrm{RG}}_{\mu,c}(T, \pi)}{\log T} \geq 2 \sum_{i>1} \frac{\Delta_i \wedge c}{\Delta_i^2}.$$

*For any abstention regret $c > 0$ and time horizon $T \geq K$, there exists a universal constant $\alpha > 0$ such that*

$$\inf_{\pi \in \Pi^{\mathrm{RG}}} \sup_{\mu \in \mathcal{U}} R^{\mathrm{RG}}_{\mu,c}(T, \pi) \geq \alpha(\sqrt{KT} \wedge cT).$$

Comparing the upper bounds on the cumulative regret of our algorithm FRG-TSwA in Theorem 1 with the corresponding lower bounds in Theorem 2, it becomes evident that our algorithm exhibits both asymptotic and minimax optimality.

**Asymptotic optimality.** For any abstention regret $c > 0$ and bandit instance $\mu \in \mathcal{U}$, the regret of our algorithm satisfies the following limiting behaviour:

$$\lim_{T \to \infty} \frac{R^{\mathrm{RG}}_{\mu,c}(T)}{\log T} = 2 \sum_{i>1} \frac{\Delta_i \wedge c}{\Delta_i^2}.$$

The above asymptotically optimal result yields several intriguing implications. First, the inclusion of the additional fixed-regret abstention option does not obviate the necessity of differentiating between suboptimal arms and the optimal one, and the exploration-exploitation trade-off remains crucial. In fact, to avoid the case in which the cumulative regret grows polynomially, the agent must still asymptotically allocate the same proportion of pulls to each suboptimal arm, as in the canonical model. This assertion is rigorously demonstrated in the proof of the lower bound (refer to Appendix C.2 for comprehensive details). Nevertheless, the abstention option does indeed reduce the exploration cost for the agent. Specifically, when exploring any suboptimal arm with a suboptimality gap larger than $c$, our algorithm leans towards employing the abstention option to minimize the instantaneous regret. This aspect is formally established in the proof of the asymptotic upper bound (see Appendix C.1 for further details).

**Minimax optimality.** In the context of worst-case guarantees for the cumulative regret, we focus on the dependence on the problem parameters: $c$, $K$ and $T$. Notably, the $\sum_{i>1} \Delta_i$ term[3] is typically considered as negligible in the literature (Audibert & Bubeck, 2009; Agrawal & Goyal, 2017; Lattimore & Szepesvári, 2020). Therefore, Theorem 1 demonstrates that our algorithm attains a worst-case regret of $O(\sqrt{KT} \wedge cT)$, which is minimax optimal in light of Theorem 2.

A phase transition phenomenon can be clearly observed from the worst-case guarantees, which dovetails with our intuitive understanding of the fixed-regret abstention setting. When the abstention regret $c$ is sufficiently low, it becomes advantageous to consistently opt for abstention to avoid the worst-case scenarios. On the contrary, when the abstention regret $c$ exceeds a certain threshold, the abstention option proves to be inadequate in alleviating the worst-case regret, as compared to the canonical model.

*Remark* 4. Although our model allows for the selected arm $A_t$ and the abstention option $B_t$ to depend on each other, the procedure used in both algorithms within this work is to first determine $A_t$ before $B_t$; this successfully achieves both forms of optimality. Nevertheless, this approach might no longer be optimal beyond the canonical $K$-armed bandit setting. In $K$-armed bandits, each arm operates independently. Conversely, in models like linear bandits, pulling one arm can indirectly reveal information about other arms. Policies based on the principle of optimism in the face of uncertainty, as well as Thompson Sampling, fall short of achieving asymptotic optimality in the context of linear bandits (Lattimore & Szepesvári, 2017). Therefore, the abstention option becomes particularly attractive if there exists an arm that incurs a substantial regret but offers significant insights into the broader bandit instance.

---

[3]This term is unavoidable when the abstention regret $c$ is sufficiently high, since every reasonable algorithm has to allocate a fixed number of pulls to each arm.

---

**Algorithm 2** Fixed-Reward Algorithm with Abstention (or FRW-ALGwA)

---

**Input:** Arm set $[K]$, abstention reward $c \in \mathbb{R}$, and a base algorithm ALG that is both asymptotically and minimax optimal for the canonical multi-armed bandit model.

 1: Initialize $\hat{\mu}_i(0) = +\infty$ for all arms $i \in [K]$.
 2: **for** $t = 1, 2, \ldots, T$ **do**
 3:    Pull the arm $A_t$ chosen by the base algorithm ALG.
 4:    Choose to abstain ($B_t = 1$) if and only if $\hat{\mu}_{A_t}(t-1) \leq c$.
 5:    Observe $X_t$ from the arm $A_t$, and update $\hat{\mu}_i(t)$ for all $i \in [K]$.
 6: **end for**

---

## 4 FIXED-REWARD SETTING

In this section, we investigate the fixed-reward setting. Here, the reward associated with the abstention option remains consistently fixed at $c \in \mathbb{R}$. When exploring a specific arm, the agent has the capability to determine whether selecting the abstention option yields a higher reward (or equivalently, a lower regret) *solely* based on its own estimated mean reward. However, in the fixed-regret setting, this decision can only be made by taking into account *both* its own estimated mean reward and the estimated mean reward of the potentially best arm. In this regard, the fixed-reward setting is inherently less complex than the fixed-regret setting. As a result, it becomes possible for us to design a more general strategy Fixed-Reward Algorithm with Abstention (or FRW-ALGwA), whose pseudocode is presented in Algorithm 2. Despite the straightforward nature of our algorithm, we demonstrate its dual attainment of both asymptotic and minimax optimality through an exhaustive theoretical examination in Sections 4.1 and 4.2.

As its name suggests, our algorithm FRW-ALGwA leverages a base algorithm ALG that is asymptotically and minimax optimal for canonical multi-armed bandits as its input. For comprehensive definitions of asymptotic and minimax optimality within the canonical model, we refer the reader to Appendix A. Notably, eligible candidate algorithms include KL-UCB$^{++}$ (Ménard & Garivier, 2017), ADA-UCB (Lattimore, 2018), MOTS-$\mathcal{J}$ (Jin et al., 2021) and Less-Exploring Thompson Sampling (Jin et al., 2023). In the operation of our algorithm, at each time step $t$, the base algorithm determines the selected arm $A_t$ according to the partial interaction historical information $(A_1, X_1, A_2, X_2, \ldots, A_{t-1}, X_{t-1})$. Subsequently, the algorithm decides whether or not to abstain, indicated by the binary random variable $B_t$, by comparing the empirical mean of the arm $A_t$, denoted as $\hat{\mu}_{A_t}(t-1)$, to the abstention reward $c$.

### 4.1 UPPER BOUNDS

Recall the definition of the cumulative regret $R_{\mu,c}^{\mathrm{RW}}(T)$, as presented in Equation (2) for the fixed-reward setting. Theorem 3 establishes both the instance-dependent asymptotic and instance-independent minimax upper bounds for Algorithm 2; see Appendix D.1 for the proof.

**Theorem 3.** *For all abstention rewards $c \in \mathbb{R}$ and bandit instances $\mu \in \mathcal{U}$, Algorithm 2 guarantees that*

$$\limsup_{T \to \infty} \frac{R_{\mu,c}^{\mathrm{RW}}(T)}{\log T} \leq 2 \sum_{i>1} \frac{\mu_1 \vee c - \mu_i \vee c}{\Delta_i^2}.$$

*Furthermore, there exists a universal constant $\alpha > 0$ such that*

$$R_{\mu,c}^{\mathrm{RW}}(T) \leq \alpha \left( \sqrt{KT} + \sum_{i \in [K]} (\mu_1 \vee c - \mu_i) \right).$$

*Remark* 5. It is worth considering the special case where $c \geq \mu_1$, where opting for abstention results in a reward even greater than, or equal to, the mean reward of the best arm. For this particular case, as per Theorem 3, since $\mu_1 \vee c - \mu_i \vee c = 0$ for all $i > 1$, our algorithm achieves a regret of $o(\log T)$. This result, in fact, is not surprising. In contrast to the fixed-regret setting where the regret associated with the abstention option is strictly positive, in this specific scenario of the fixed-reward setting, selecting the abstention option is indeed the optimal action at a single time step, regardless of the arm pulled. Therefore, there is no necessity to distinguish between suboptimal arms and the optimal one, and the

exploration-exploitation trade-off becomes inconsequential. However, when the abstention reward is below the mean reward of the best arm, i.e., $c < \mu_1$, maintaining a subpolynomial cumulative regret still hinges on the delicate balance between exploration and exploitation, as evidenced by the forthcoming exposition of the asymptotic lower bound.

## 4.2 LOWER BOUNDS

We hereby introduce the concept of $R^{\mathrm{RW}}$-consistency for the fixed-reward setting, in a manner analogous to the fixed-regret setting. Following this, we present two distinct lower bounds for the problem of regret minimization in multi-armed bandits with fixed-reward abstention in Theorem 4. The proof for Theorem 4 is postponed to Appendix D.2.

**Definition 2** ($R^{\mathrm{RW}}$-consistency)**.** An algorithm $\pi \in \Pi^{\mathrm{RW}}$ is said to be $R^{\mathrm{RW}}$-*consistent* if for all abstention rewards $c \in \mathbb{R}$, bandit instances $\mu \in \mathcal{U}$, and $a > 0$, $R^{\mathrm{RW}}_{\mu,c}(T, \pi) = o(T^a)$.

**Theorem 4.** *For any abstention reward $c \in \mathbb{R}$, bandit instance $\mu \in \mathcal{U}$ and $R^{\mathrm{RW}}$-consistent algorithm $\pi$, it holds that*

$$\liminf_{T \to \infty} \frac{R^{\mathrm{RW}}_{\mu,c}(T, \pi)}{\log T} \geq 2 \sum_{i>1} \frac{\mu_1 \vee c - \mu_i \vee c}{\Delta_i^2}.$$

*For any abstention reward $c \in \mathbb{R}$ and time horizon $T \geq K$, there exists a universal constant $\alpha > 0$ such that*

$$\inf_{\pi \in \Pi^{\mathrm{RW}}} \sup_{\mu \in \mathcal{U}} R^{\mathrm{RW}}_{\mu,c}(T, \pi) \geq \alpha \sqrt{KT}.$$

By comparing the upper bounds in Theorem 3 with the lower bounds in Theorem 4, it is firmly confirmed that Algorithm 2 is both asymptotically and minimax optimal in the fixed-reward setting.

**Asymptotic optimality.** For any abstention reward $c \in \mathbb{R}$ and bandit instance $\mu \in \mathcal{U}$, our algorithm ensures the following optimal asymptotic behavior for the cumulative regret:

$$\lim_{T \to \infty} \frac{R^{\mathrm{RW}}_{\mu,c}(T)}{\log T} = 2 \sum_{i>1} \frac{\mu_1 \vee c - \mu_i \vee c}{\Delta_i^2}.$$

Since it holds generally that $\mu_1 \vee c - \mu_i \vee c \leq \Delta_i$ for each arm $i > 1$, our algorithm effectively reduces the cumulative regret in the asymptotic regime through the incorporation of the fixed-reward abstention option.

**Minimax optimality.** As for the worst-case performance of our algorithm, disregarding the additive term $\sum_{i \in [K]} (\mu_1 \vee c - \mu_i)$, it achieves an optimal worst-case regret of $O(\sqrt{KT})$. While this worst-case regret aligns with that in the canonical multi-armed bandit model, it is noteworthy that this achievement is non-trivial, demanding meticulous management of the asymptotic regret performance in parallel.

Moreover, there is no occurrence of the phase transition phenomenon in the fixed-reward setting. This absence can be attributed to the intrinsic nature of the fixed-reward abstention option. For any abstention reward $c \in \mathbb{R}$ and online algorithm, we can always construct a challenging bandit instance that leads to a cumulative regret of $\Omega(\sqrt{KT})$, as demonstrated in the proof of the minimax lower bound in Appendix D.2.

## 5 NUMERICAL EXPERIMENTS

In this section, we conduct numerical experiments to empirically validate our theoretical insights. Due to space limitations, we report our results only for the fixed-regret setting here. Results pertaining to the fixed-reward setting are available in Appendix E. In each experiment, the reported cumulative regrets are averaged over $2,000$ independent trials and the corresponding standard deviations are displayed as error bars in the figures.

To confirm the benefits of incorporating the abstention option, we compare the performance of our proposed algorithm FRG-TSwA (Algorithm 1) with that of Less-Exploring Thompson Sampling

(Jin et al., 2023), which serves as a baseline algorithm without the abstention option. We consider two synthetic bandit instances. The first instance $\mu^{\dagger}$ with $K = 7$ has uniform suboptimality gaps: $\mu_1^{\dagger} = 1$ and $\mu_i^{\dagger} = 0.7$ for all $i \in [K] \setminus \{1\}$. For the second instance $\mu^{\ddagger}$ with $K = 10$, the suboptimality gaps are more diverse: $\mu_1^{\ddagger} = 1$, $\mu_i^{\ddagger} = 0.7$ for $i \in \{2, 3, 4\}$, $\mu_i^{\ddagger} = 0.5$ for $i \in \{5, 6, 7\}$ and $\mu_i^{\ddagger} = 0.3$ for $i \in \{8, 9, 10\}$. The empirical averaged cumulative regrets of both methods with

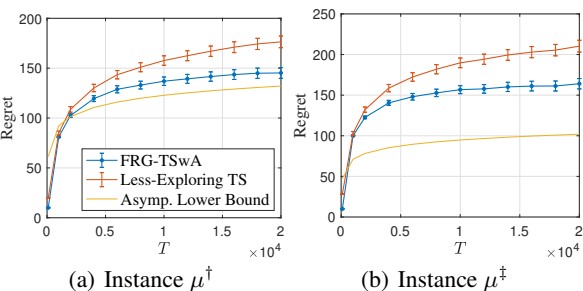

(a) Instance $\mu^{\dagger}$      (b) Instance $\mu^{\ddagger}$

Figure 2: Empirical regrets with abstention regret $c = 0.1$ for different time horizons $T$.

abstention regret $c = 0.1$ for different time horizons $T$ are presented in Figure 2. To demonstrate their asymptotic behavior, we also plot the instance-dependent asymptotic lower bound on the cumulative regret (see Theorem 2) in each sub-figure. It can be observed that FRG-TSwA is clearly superior compared to the non-abstaining baseline, especially for large values of $T$. This demonstrates the advantage of the abstention mechanism. With regard to the growth trend, as the time horizon $T$ increases, the curve corresponding to FRG-TSwA closely approximates that of the asymptotic lower bound. This suggests that the expected cumulative regret of FRG-TSwA matches the lower bound asymptotically, thereby substantiating the theoretical results presented in Section 3.

To illustrate the effect of the abstention regret $c$, we evaluate the performance of FRG-TSwA for varying values of $c$, while keeping the time horizon $T$ fixed at $10,000$. The experimental results for both bandit instances $\mu^{\dagger}$ and $\mu^{\ddagger}$ are presented in Figure 3. Within each sub-figure, we observe that as $c$ increases, the empirical averaged cumulative regret initially increases but eventually saturates beyond a certain threshold value of $c$. These empirical observations align well with our expectations. Indeed, when provided with com-

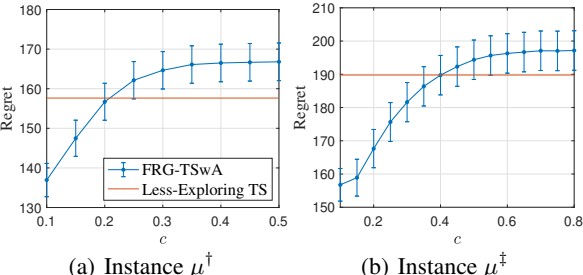

(a) Instance $\mu^{\dagger}$      (b) Instance $\mu^{\ddagger}$

Figure 3: Empirical regrets with time horizon $T = 10,000$ for different abstention regrets $c$.

plete information about the bandit instance, if the abstention regret $c$ exceeds the largest suboptimality gap, the agent gains no advantage in choosing the abstention option when selecting any arm. However, we remark that the agent lacks this oracle-like knowledge of the suboptimality gaps and must estimate them on the fly. Consequently, this results in the inevitable selection of the abstention option, even when the abstention regret $c$ is large.

## 6   CONCLUSIONS AND FUTURE WORK

In this paper, we consider, for the first time, a multi-armed bandit model that allows for the possibility of *abstaining* from accepting the stochastic rewards, alongside the conventional arm selection. This innovative framework is motivated by real-world scenarios where decision-makers may wish to hedge against highly uncertain or risky actions, as exemplified in clinical trials. Within this enriched paradigm, we address both the fixed-regret and fixed-reward settings, providing tight upper and lower bounds on asymptotic and minimax regrets for each scenario. For the fixed-regret setting, we thoughtfully adapt a recently developed asymptotically and minimax optimal algorithm by Jin et al. (2023) to accommodate the abstention option while preserving its attractive optimality characteristics. For the fixed-reward setting, we convert *any* asymptotically and minimax optimal algorithm for the canonical model into one that retains these optimality properties when the abstention option is present. Finally, experiments on synthetic datasets validate our theoretical results and clearly demonstrate the advantage of incorporating the abstention option.

As highlighted in Remark 4, a fruitful avenue for future research lies in expanding the abstention model from $K$-armed bandits to linear bandits. An intriguing inquiry is whether the inclusion of the abstention feature can lead to enhanced asymptotic and minimax theoretical guarantees.

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

# A ASYMPTOTIC AND MINIMAX OPTIMALITY IN CANONICAL MULTI-ARMED BANDITS

In the canonical multi-armed bandit model, there is no additional abstention option. Given a bandit instance $\mu \in \mathcal{U}$, at each time step $t \in \mathbb{N}$, the agent employs an *online algorithm* $\pi$ to choose an arm $A_t$ from the arm set $[K]$, and then observes a random variable $X_t$ from the selected arm $A_t$, which is drawn from a Gaussian distribution $\mathcal{N}(\mu_{A_t}, 1)$ and independent of observations from previous time steps. The choice of $A_t$ might depend on the prior decisions and observations. To describe the setup formally, $A_t$ is $\mathcal{F}_{t-1}^{\mathrm{CA}}$-measurable, where $\mathcal{F}_t^{\mathrm{CA}} := \sigma(A_1, X_1, A_2, X_2, \ldots, A_t, X_t)$ represents the $\sigma$-field generated by the cumulative interaction history up to and including time $t$. Subsequently, the agent suffers an instantaneous regret of $\mu_1 - X_t$.

The agent aims at minimizing the expected cumulative regret over a time horizon $T$, which is defined as

$$R_\mu^{\mathrm{CA}}(T, \pi) = T\mu_1 - \mathbb{E}\left[\sum_{t=1}^{T} X_t\right].$$

We refer to the collection of all online policies for the canonical multi-armed bandit model as $\Pi^{\mathrm{CA}}$.

*Remark* 6. It is worth noting that any algorithm designed for canonical multi-armed bandit model possesses the capability to decide the arm $A_t$ to pull at each time step $t$, based on the partial interaction history $(A_1, X_1, A_2, X_2, \ldots, A_{t-1}, X_{t-1})$, within the abstention model. Conversely, any algorithm tailored for the abstention model in the fixed-regret setting (or in the fixed-reward setting) can be applied to the canonical multi-armed bandit model, provided that the abstention regret (or the abstention reward) has been predetermined. Specifically, the algorithm can determine both the selected arm $A_t$ and the binary abstention variable $B_t$, although $B_t$ is purely auxiliary and exerts no influence on the cumulative regret $R_\mu^{\mathrm{CA}}(T, \pi)$.

**Lower bounds.** Both the asymptotic and minimax lower bounds for the canonical multi-armed bandit model have been thoroughly established (Lai & Robbins, 1985; Auer et al., 1995). For a comprehensive overview, refer to Sections 15 and 16 of Lattimore & Szepesvári (2020). Here, we summarize the results in the following:

**Definition 3** ($R^{\mathrm{CA}}$-consistency). An algorithm $\pi \in \Pi^{\mathrm{CA}}$ is said to be $R^{\mathrm{CA}}$-*consistent* if for all bandit instances $\mu \in \mathcal{U}$ and $a > 0$, $R_\mu^{\mathrm{CA}}(T, \pi) = o(T^a)$.

**Theorem 5.** *For any bandit instance $\mu \in \mathcal{U}$ and $R^{\mathrm{CA}}$-consistent algorithm $\pi$, it holds that*

$$\liminf_{T \to \infty} \frac{R_\mu^{\mathrm{CA}}(T, \pi)}{\log T} \geq \sum_{i>1} \frac{2}{\Delta_i}.$$

*For any time horizon $T \geq K$, there exists a universal constant $\alpha > 0$ such that*

$$\inf_{\pi \in \Pi^{\mathrm{CA}}} \sup_{\mu \in \mathcal{U}} R_\mu^{\mathrm{CA}}(T, \pi) \geq \alpha\sqrt{KT}.$$

**Asymptotic and minimax optimality.** According to Theorem 5, in the canonical bandit model, an algorithm $\pi \in \Pi^{\mathrm{CA}}$ is said to be *asymptotically optimal* if for all bandit instances $\mu \in \mathcal{U}$, it ensures that

$$\lim_{T \to \infty} \frac{R_\mu^{\mathrm{CA}}(T, \pi)}{\log T} = \sum_{i>1} \frac{2}{\Delta_i}.$$

Furthermore, it is said to be *minimax optimal* if there exists a universal constant $\alpha > 0$ such that

$$R_\mu^{\mathrm{CA}}(T, \pi) \leq \alpha\left(\sqrt{KT} + \sum_{i>1} \Delta_i\right).$$

To the best of our knowledge, for canonical multi-armed bandits with Gaussian rewards, KL-UCB$^{++}$ (Ménard & Garivier, 2017), ADA-UCB (Lattimore, 2018), MOTS-$\mathcal{J}$ (Jin et al., 2021)

and Less-Exploring Thompson Sampling (Jin et al., 2023) exhibit simultaneous asymptotic and minimax optimality. As their names suggest, the former two algorithms follow the UCB-style, while the latter two are rooted in Thompson Sampling.

*Remark* 7. One valuable byproduct derived from the proof of the asymptotic lower bound in Theorem 5 is that, for any $R^{\mathrm{CA}}$-consistent algorithm $\pi$, bandit instance $\mu \in \mathcal{U}$ and suboptimal arm $i > 1$, we have

$$\liminf_{T \to \infty} \frac{\mathbb{E}[N_i(T)]}{\log T} \geq \frac{2}{\Delta_i^2}.$$

Therefore, any algorithm that is asymptotically optimal ensures that for all suboptimal arms $i > 1$,

$$\lim_{T \to \infty} \frac{\mathbb{E}[N_i(T)]}{\log T} = \frac{2}{\Delta_i^2}.$$

## B  AUXILIARY LEMMAS

**Lemma 1** (Bretagnolle–Huber inequality (Tsybakov, 2009)). *Let $\mathbb{P}$ and $\mathbb{P}'$ be two probability distributions on the same measurable space $(\Omega, \mathcal{F})$. For any event $A \in \mathcal{F}$ and its complement $A^c = \Omega \backslash A$, the following inequality holds:*

$$\mathbb{P}(A) + \mathbb{P}'(A^c) \geq \frac{1}{2} \exp(-\mathrm{KL}(\mathbb{P}, \mathbb{P}')),$$

*where $\mathrm{KL}(\mathbb{P}, \mathbb{P}')$ denotes the Kullback–Leibler (KL) divergence between $\mathbb{P}$ and $\mathbb{P}'$.*

**Lemma 2** (Divergence decomposition lemma). *Consider both the fixed-regret setting and the fixed-reward setting. Fix an arbitrary policy $\pi$. Let $\nu = (\mathbb{P}_1, \dots, \mathbb{P}_K)$ represent the reward distributions associated with one bandit instance, and let $\nu' = (\mathbb{P}'_1, \dots, \mathbb{P}'_K)$ represent the reward distributions associated with another bandit instance. Define $\mathbb{P}_{\nu,c}$ as the probability distribution of the sequence $(A_1, B_1, X_1, \dots, A_T, B_T, X_T)$ induced by the algorithm $\pi$ under the abstention regret $c$ in the fixed-regret setting (or the abstention reward $c$ in the fixed-reward setting) for the bandit instance $\nu$. Similarly, let $\mathbb{P}_{\nu',c}$ denote the same for the bandit instance $\nu'$. Then the KL divergence between $\mathbb{P}_{\nu,c}$ and $\mathbb{P}_{\nu',c}$ can be decomposed as:*

$$\mathrm{KL}\left(\mathbb{P}_{\nu,c}, \mathbb{P}_{\nu',c}\right) = \sum_{i \in [K]} \mathbb{E}_{\nu,c}[N_i(T)] \mathrm{KL}\left(\mathbb{P}_i, \mathbb{P}'_i\right).$$

The proof of Lemma 2 is similar to the well-known proof of divergence decomposition in the canonical multi-armed bandit model (excluding abstention), and is therefore omitted. This proof can be located, for instance, in Garivier et al. (2019, Section 2.1) and Lattimore & Szepesvári (2020, Lemma 15.1).

**Lemma 3** (Hoeffding's inequality for sub-Gaussian random variables). *Let $X_1, \dots, X_n$ be independent $\sigma$-sub-Gaussian random variables with mean $\mu$. Then for any $\varepsilon \geq 0$,*

$$\mathbb{P}(\hat{\mu} \geq \mu + \varepsilon) \leq \exp\left(-\frac{n\varepsilon^2}{2\sigma^2}\right) \quad and \quad \mathbb{P}(\hat{\mu} \leq \mu - \varepsilon) \leq \exp\left(-\frac{n\varepsilon^2}{2\sigma^2}\right)$$

*where $\hat{\mu} := \frac{1}{n} \sum_{i=1}^n X_i$.*

**Lemma 4.** *Let $\{X_i\}_{i \in \mathbb{N}}$ be a sequence of independent $\sigma$-sub-Gaussian random variables with mean $\mu$. Then for any $\varepsilon > 0$ and $N \in \mathbb{N}$,*

$$\sum_{n=1}^N \mathbb{P}(\hat{\mu}_n \geq \mu + \varepsilon) \leq \frac{2\sigma^2}{\varepsilon^2} \quad and \quad \sum_{n=1}^N \mathbb{P}(\hat{\mu}_n \leq \mu - \varepsilon) \leq \frac{2\sigma^2}{\varepsilon^2}$$

*where $\hat{\mu}_n := \frac{1}{n} \sum_{i=1}^n X_i$.*

*Proof of Lemma 4.* By symmetry, it suffices to prove the first part. According to Lemma 3, we have

$$
\begin{aligned}
\sum_{n=1}^{N} \mathbb{P}(\hat{\mu}_n \geq \mu + \varepsilon) &\leq \sum_{n=1}^{N} \exp\left(-\frac{n\varepsilon^2}{2\sigma^2}\right) \\
&\leq \frac{\exp\left(-\frac{\varepsilon^2}{2\sigma^2}\right)}{1 - \exp\left(-\frac{\varepsilon^2}{2\sigma^2}\right)} \\
&= \frac{1}{\exp\left(\frac{\varepsilon^2}{2\sigma^2}\right) - 1} \\
&\leq \frac{2\sigma^2}{\varepsilon^2}
\end{aligned}
$$

where the last inequality follows from the fact that $e^x - 1 \geq x$ for any $x \geq 0$. $\qquad\square$

## C ANALYSIS OF THE FIXED-REGRET SETTING

### C.1 UPPER BOUNDS

*Proof of Theorem 1.* Due to the law of total expectation, we can decompose the regret $R_{\mu,c}^{\mathrm{RG}}(T, \pi)$ as

$$
\begin{aligned}
R_{\mu,c}^{\mathrm{RG}}(T, \pi) &= \mathbb{E}\left[\sum_{t=1}^{T} \left((\mu_1 - X_t) \cdot \mathbb{1}\{B_t = 0\} + c \cdot \mathbb{1}\{B_t = 1\}\right)\right] \\
&= \mathbb{E}\left[\sum_{t=1}^{T} \left((\mu_1 - \mu_{A_t}) \cdot \mathbb{1}\{B_t = 0\} + c \cdot \mathbb{1}\{B_t = 1\}\right)\right] \\
&= c \cdot \mathbb{E}[N_1^{(1)}(T)] + \sum_{i>1} \left(\Delta_i \cdot \mathbb{E}[N_i^{(0)}(T)] + c \cdot \mathbb{E}[N_i^{(1)}(T)]\right). \quad (3)
\end{aligned}
$$

For any arm $i$ with $\Delta_i < c$ (including the best arm), it holds that

$$
\begin{aligned}
&\mathbb{E}[N_i^{(1)}(T)] \\
&= \mathbb{E}\left[\sum_{t=1}^{T} \mathbb{I}\{A_t = i \text{ and } B_t = 1\}\right] \\
&\leq \mathbb{E}\left[\sum_{t=1}^{T} \mathbb{I}\left\{A_t = i \text{ and } \sqrt{\frac{K}{t}} \geq c\right\}\right] \\
&\quad + \mathbb{E}\left[\sum_{t=K+1}^{T} \mathbb{I}\left\{A_t = i \text{ and } \max_{j \in [K] \setminus \{i\}} \left(\hat{\mu}_j(t-1) - \sqrt{\frac{6 \log t + 2 \log(c \vee 1)}{N_j(t-1)}}\right) - \hat{\mu}_i(t-1) \geq c\right\}\right] \\
&\leq \mathbb{E}\left[\sum_{t=1}^{T} \mathbb{I}\left\{A_t = i \text{ and } \sqrt{\frac{K}{t}} \geq c\right\}\right] \\
&\quad + \mathbb{E}\left[\sum_{t=K+1}^{T} \mathbb{I}\left\{A_t = i \text{ and } \max_{j \in [K]} \left(\hat{\mu}_j(t-1) - \sqrt{\frac{6 \log t + 2 \log(c \vee 1)}{N_j(t-1)}}\right) \geq \mu_1\right\}\right] \\
&\quad + \mathbb{E}\left[\sum_{t=K+1}^{T} \mathbb{I}\{A_t = i \text{ and } \mu_1 - \hat{\mu}_i(t-1) \geq c\}\right] \quad (4)
\end{aligned}
$$

where the last inequality arises from the observation that when $A_t = i$,

$$\left\{ \max_{j \in [K] \setminus \{i\}} \left( \hat{\mu}_j(t-1) - \sqrt{\frac{6 \log t + 2 \log(c \vee 1)}{N_j(t-1)}} \right) - \hat{\mu}_i(t-1) \geq c \right\}$$

$$\subseteq \left\{ \max_{j \in [K] \setminus \{i\}} \left( \hat{\mu}_j(t-1) - \sqrt{\frac{6 \log t + 2 \log(c \vee 1)}{N_j(t-1)}} \right) \geq \mu_1 \right\} \cup \{\mu_1 - \hat{\mu}_i(t-1) \geq c\}$$

$$\subseteq \left\{ \max_{j \in [K]} \left( \hat{\mu}_j(t-1) - \sqrt{\frac{6 \log t + 2 \log(c \vee 1)}{N_j(t-1)}} \right) \geq \mu_1 \right\} \cup \{\mu_1 - \hat{\mu}_i(t-1) \geq c\}.$$

For convenience, for any $i \in [K]$ such that $\Delta_i < c$, we introduce three shorthand notations to represent the terms in (4):

$$\begin{cases} (\clubsuit)_i := \mathbb{E}\left[ \sum_{t=1}^T \mathbb{I}\left\{ A_t = i \text{ and } \sqrt{\frac{K}{t}} \geq c \right\} \right] \\ (\spadesuit)_i := \mathbb{E}\left[ \sum_{t=K+1}^T \mathbb{I}\left\{ A_t = i \text{ and } \max_{j \in [K]} \left( \hat{\mu}_j(t-1) - \sqrt{\frac{6 \log t + 2 \log(c \vee 1)}{N_j(t-1)}} \right) \geq \mu_1 \right\} \right] \\ (\blacksquare)_i := \mathbb{E}\left[ \sum_{t=K+1}^T \mathbb{I}\{ A_t = i \text{ and } \mu_1 - \hat{\mu}_i(t-1) \geq c \} \right]. \end{cases}$$

We will deal with $(\clubsuit)_i$ and $(\spadesuit)_i$ later for the two forms of upper bounds.

On the other hand, for the term $(\blacksquare)_i$, we have

$$(\blacksquare)_i \leq \mathbb{E}\left[ \sum_{t=K+1}^T \sum_{s=1}^{T-1} \mathbb{I}\{ A_t = i \text{ and } \hat{\mu}_{is} \leq \mu_i - (c - \Delta_i) \text{ and } N_i(t-1) = s \} \right]$$

$$\leq \mathbb{E}\left[ \sum_{s=1}^{T-1} \mathbb{I}\{ \hat{\mu}_{is} \leq \mu_i - (c - \Delta_i) \} \right] \tag{5}$$

$$\leq \frac{2}{(c - \Delta_i)^2}. \tag{6}$$

Line (5) follows from the fact that for all $s \in [T-1]$,

$$\sum_{t=K+1}^T \mathbb{I}\{ A_t = i \text{ and } N_i(t-1) = s \} \leq 1.$$

Line (6) is due to Lemma 4.

For any arm $i$ with $\Delta_i > c$, since arm $1 \in [K] \setminus \{i\}$, we have

$$\mathbb{E}[N_i^{(0)}(T)]$$

$$= \mathbb{E}\left[ \sum_{t=1}^T \mathbb{I}\{ A_t = i \text{ and } B_t = 0 \} \right]$$

$$\leq 1 + \mathbb{E}\left[ \sum_{t=K+1}^T \mathbb{I}\left\{ A_t = i \text{ and } \right.\right.$$

$$\left.\left. \max_{j \in [K] \setminus \{i\}} \left( \hat{\mu}_j(t-1) - \sqrt{\frac{6 \log t + 2 \log(c \vee 1)}{N_j(t-1)}} \right) - \hat{\mu}_i(t-1) < c \text{ and } \sqrt{\frac{K}{t}} < c \right\} \right]$$

$$\leq 1 + \mathbb{E}\left[ \sum_{t=K+1}^T \mathbb{I}\left\{ A_t = i \text{ and } \left( \hat{\mu}_1(t-1) - \sqrt{\frac{6 \log t + 2 \log(c \vee 1)}{N_1(t-1)}} \right) - \hat{\mu}_i(t-1) < c \right\} \right]. \tag{7}$$

**Minimax upper bound.** If $c \leq \sqrt{\frac{K}{T}}$, then the abstention option is always invoked because $\sqrt{\frac{K}{t}} \geq \sqrt{\frac{K}{T}} \geq c$ for all $t \in [T]$. Consequently, it is straightforward to deduce that

$$R_{\mu,c}^{\mathrm{RG}}(T) \leq cT.$$

Next, consider the case that $c > \sqrt{\frac{K}{T}}$. Compared with the canonical multi-armed bandit model, at a single time step, the agent in our abstention model incurs a greater (expected) regret only if an arm $i$ with $\Delta_i < c$ is pulled and the abstention option is selected. Thus, we have

$$R_{\mu,c}^{\mathrm{RG}}(T) \leq R_{\mu}^{\mathrm{CA}}(T) + \sum_{i:\Delta_i < c} (c - \Delta_i) \cdot \mathbb{E}[N_i^{(1)}(T)]. \tag{8}$$

Due to the minimax optimality of Less-Exploring Thompson Sampling (Jin et al., 2023), there exists a universal constant $\alpha_1 > 0$ such that

$$R_{\mu}^{\mathrm{CA}}(T) \leq \alpha_1 \left( \sqrt{KT} + \sum_{i>1} \Delta_i \right). \tag{9}$$

Recall the upper bound of $\mathbb{E}[N_i^{(1)}(T)]$ for arm $i$ with $\Delta_i < c$, as given in (4). Subsequently, we will establish bounds for the following terms:

$$\sum_{i:\Delta_i < c} (c - \Delta_i) \cdot (\clubsuit)_i \,, \quad \sum_{i:\Delta_i < c} (c - \Delta_i) \cdot (\spadesuit)_i \quad \text{and} \quad \sum_{i:\Delta_i < c} (c - \Delta_i) \cdot ((\blacksquare)_i.$$

For the first term, we have

$$\begin{aligned}
\sum_{i:\Delta_i < c} (c - \Delta_i) \cdot (\clubsuit)_i &\leq \sum_{i:\Delta_i < c} c \cdot \mathbb{E}\left[ \sum_{t=1}^{T} \mathbb{I}\left\{ A_t = i \text{ and } \sqrt{\frac{K}{t}} \geq c \right\} \right] \\
&\leq c \cdot \mathbb{E}\left[ \sum_{t=1}^{T} \mathbb{I}\left\{ \sqrt{\frac{K}{t}} \geq c \right\} \right] \\
&\leq \frac{K}{c} \\
&\leq \sqrt{KT}. \tag{10}
\end{aligned}$$

For the second term, we can obtain

$$\begin{aligned}
\sum_{i:\Delta_i < c} (c - \Delta_i) \cdot (\spadesuit)_i &\leq \sum_{i:\Delta_i < c} c \cdot (\spadesuit)_i \\
&\leq c \cdot \mathbb{E}\left[ \sum_{t=K+1}^{T} \mathbb{I}\left\{ \max_{j \in [K]} \left( \hat{\mu}_j(t-1) - \sqrt{\frac{6\log t + 2\log(c \vee 1)}{N_j(t-1)}} \right) \geq \mu_1 \right\} \right] \\
&\leq c \cdot \sum_{j \in [K]} \mathbb{E}\left[ \sum_{t=K+1}^{T} \mathbb{I}\left\{ \hat{\mu}_j(t-1) - \sqrt{\frac{6\log t + 2\log(c \vee 1)}{N_j(t-1)}} \geq \mu_1 \right\} \right].
\end{aligned}$$

For all $j \in [K]$, by a union bound over all possible values of $N_j(t-1)$ and Lemma 3, we have

$$\mathbb{E}\left[\sum_{t=K+1}^{T} \mathbb{I}\left\{\hat{\mu}_j(t-1) - \sqrt{\frac{6\log t + 2\log(c \vee 1)}{N_j(t-1)}} \geq \mu_1\right\}\right]$$

$$\leq \sum_{t=K+1}^{T} \sum_{s=1}^{t-1} \mathbb{P}\left(\hat{\mu}_{js} - \sqrt{\frac{6\log t + 2\log(c \vee 1)}{s}} \geq \mu_1\right)$$

$$\leq \sum_{t=K+1}^{T} \sum_{s=1}^{t-1} \mathbb{P}\left(\hat{\mu}_{js} - \sqrt{\frac{6\log t + 2\log(c \vee 1)}{s}} \geq \mu_j\right) \tag{11}$$

$$\leq \sum_{t=K+1}^{T} \sum_{s=1}^{t-1} \frac{1}{t^3(c \vee 1)}$$

$$\leq \sum_{t=K+1}^{T} \frac{1}{t^2 c}$$

$$\leq \frac{1}{Kc}$$

where the last inequality follows from the numerical fact that

$$\sum_{t=K+1}^{T} \frac{1}{t^2} \leq \int_{x=K}^{\infty} \frac{1}{x^2}\, \mathrm{d}x = \frac{1}{K}.$$

Thus, we can bound the second term as

$$\sum_{i:\Delta_i < c} (c - \Delta_i) \cdot (\spadesuit)_i \leq c \cdot \sum_{j \in [K]} \frac{1}{Kc} = 1. \tag{12}$$

For the third term, in addition to the upper bound of $(\blacksquare)_i$ in (6), we can identify another straightforward upper bound as follows:

$$(\blacksquare)_i \leq \mathbb{E}\left[\sum_{t=K+1}^{T} \mathbb{I}\{A_t = i\}\right] \leq \mathbb{E}[N_i(T)].$$

By applying these two bounds separately for distinct scenarios,, we have

$$\sum_{i:\Delta_i < c} (c - \Delta_i) \cdot (\blacksquare)_i \leq \sum_{i:0 < c - \Delta_i < \sqrt{\frac{K}{T}}} (c - \Delta_i) \cdot (\blacksquare)_i + \sum_{i:c - \Delta_i \geq \sqrt{\frac{K}{T}}} (c - \Delta_i) \cdot (\blacksquare)_i$$

$$\leq \sqrt{\frac{K}{T}} \sum_{i:0 < c - \Delta_i < \sqrt{\frac{K}{T}}} \mathbb{E}[N_i(T)] + \sum_{i:c - \Delta_i \geq \sqrt{\frac{K}{T}}} \frac{2}{c - \Delta_i}$$

$$\leq \sqrt{\frac{K}{T}} \cdot T + \sum_{i:c - \Delta_i \geq \sqrt{\frac{K}{T}}} 2\sqrt{\frac{T}{K}}$$

$$\leq \sqrt{KT} + K \cdot 2\sqrt{\frac{T}{K}}$$

$$= 3\sqrt{KT}. \tag{13}$$

By plugging Inequalities (10), (12) and (13) into (4), we have

$$\sum_{i:\Delta_i < c} (c - \Delta_i) \cdot \mathbb{E}[N_i^{(1)}(T)] \leq 1 + 4\sqrt{KT}.$$

Together with (8) and (9), we can conclude that

$$R_{\mu,c}^{\mathrm{RG}}(T) \leq (\alpha_1 + 4)\sqrt{KT} + \alpha_1 \sum_{i>1} \Delta_i + 1.$$

Therefore, there must exist a universal constant $\alpha > 0$ such that

$$R_{\mu,c}^{\mathrm{RG}}(T) \leq \alpha \left( \sqrt{KT} + \sum_{i>1} \Delta_i \right).$$

This completes the proof of the minimax upper bound.

**Asymptotic upper bound.** Consider any arm $i$ with $\Delta_i < c$ (including the best arm). In the following, we will further elucidate the upper bound of $\mathbb{E}[N_i^{(1)}(T)]$ as given in (4) within the asymptotic domain.

For the first term $(\clubsuit)_i$ in (4), we have

$$(\clubsuit)_i \leq \mathbb{E}\left[ \sum_{t=1}^{T} \mathbb{I}\left\{ \sqrt{\frac{K}{t}} \geq c \right\} \right] \leq \frac{K}{c^2}.$$

For the second term $(\spadesuit)_i$, using Inequality (11), we can get

$$\begin{aligned}
(\spadesuit)_i &\leq \mathbb{E}\left[ \sum_{t=K+1}^{T} \mathbb{I}\left\{ \max_{j \in [K]} \left( \hat{\mu}_j(t-1) - \sqrt{\frac{6\log t + 2\log(c \vee 1)}{N_j(t-1)}} \right) \geq \mu_1 \right\} \right] \\
&\leq \sum_{j \in [K]} \mathbb{E}\left[ \sum_{t=K+1}^{T} \mathbb{I}\left\{ \hat{\mu}_j(t-1) - \sqrt{\frac{6\log t + 2\log(c \vee 1)}{N_j(t-1)}} \geq \mu_1 \right\} \right] \\
&\leq \frac{1}{c}.
\end{aligned}$$

Incorporating (6), we obtain

$$\mathbb{E}[N_i^{(1)}(T)] \leq \frac{K}{c^2} + \frac{1}{c} + \frac{2}{(c-\Delta_i)^2} = o(\log T).$$

Consider any arm $i$ with $\Delta_i > c$. We will further explore the upper bound of $\mathbb{E}[N_i^{(0)}(T)]$ in (7).
According to the fact that

$$\begin{aligned}
&\left\{ A_t = i \text{ and } \left( \hat{\mu}_1(t-1) - \sqrt{\frac{6\log t + 2\log(c \vee 1)}{N_1(t-1)}} \right) - \hat{\mu}_i(t-1) < c \right\} \\
\subseteq{}& \left\{ \hat{\mu}_1(t-1) + \sqrt{\frac{6\log t + 2\log(c \vee 1)}{N_1(t-1)}} \leq \mu_1 \right\} \\
&\cup \left\{ A_t = i \text{ and } \left( \mu_1 - 2\sqrt{\frac{6\log t + 2\log(c \vee 1)}{N_1(t-1)}} \right) - \hat{\mu}_i(t-1) \leq c \right\},
\end{aligned}$$

we have

$$\mathbb{E}[N_i^{(0)}(T)]$$

$$\leq 1 + \mathbb{E}\left[\sum_{t=K+1}^{T} \mathbb{I}\left\{A_t = i \text{ and } \left(\hat{\mu}_1(t-1) - \sqrt{\frac{6\log t + 2\log(c \vee 1)}{N_1(t-1)}}\right) - \hat{\mu}_i(t-1) < c\right\}\right]$$

$$\leq 1 + \mathbb{E}\left[\sum_{t=K+1}^{T} \mathbb{I}\left\{\hat{\mu}_1(t-1) + \sqrt{\frac{6\log t + 2\log(c \vee 1)}{N_1(t-1)}} \leq \mu_1\right\}\right]$$

$$+ \underbrace{\mathbb{E}\left[\sum_{t=K+1}^{T} \mathbb{I}\left\{A_t = i \text{ and } \left(\mu_1 - 2\sqrt{\frac{6\log t + 2\log(c \vee 1)}{N_1(t-1)}}\right) - \hat{\mu}_i(t-1) < c\right\}\right]}_{(\bigstar)_i}. \qquad (14)$$

Following a similar argument as in (11), we can derive

$$\mathbb{E}\left[\sum_{t=K+1}^{T} \mathbb{I}\left\{\hat{\mu}_1(t-1) + \sqrt{\frac{6\log t + 2\log(c \vee 1)}{N_1(t-1)}} \leq \mu_1\right\}\right] \leq \frac{1}{Kc}. \qquad (15)$$

Now we focus on the last term in (14), which is denoted by $(\bigstar)_i$.

For any fixed $b \in (0,1)$, there must exist a constant $t_1(b, \mu, c) \geq K+1$ such that for all $t \geq t_1$,

$$2\sqrt{\frac{6\log t + 2\log(c \vee 1)}{(t-1)^b}} \leq \frac{\Delta_i - c}{2}.$$

Notice that for all $t \geq t_1$,

$$\left\{A_t = i \text{ and } \left(\mu_1 - 2\sqrt{\frac{6\log t + 2\log(c \vee 1)}{N_1(t-1)}}\right) - \hat{\mu}_i(t-1) < c\right\}$$

$$\subseteq \left\{N_1(t-1) \leq (t-1)^b\right\}$$

$$\cup \left\{A_t = i \text{ and } \left(\mu_1 - 2\sqrt{\frac{6\log t + 2\log(c \vee 1)}{N_1(t-1)}}\right) - \hat{\mu}_i(t-1) < c \text{ and } N_1(t-1) > (t-1)^b\right\}$$

$$\subseteq \left\{N_1(t-1) \leq (t-1)^b\right\} \cup \left\{A_t = i \text{ and } \hat{\mu}_i(t-1) \geq \mu_1 + \frac{\Delta_i - c}{2}\right\}.$$

From the above, we deduce that

$$(\bigstar)_i \leq t_1 + \sum_{t=t_1}^{T} \mathbb{P}\left(N_1(t-1) \leq (t-1)^b\right) + \mathbb{E}\left[\sum_{t=t_1}^{T} \mathbb{I}\left\{A_t = i \text{ and } \hat{\mu}_i(t-1) \geq \mu_1 + \frac{\Delta_i - c}{2}\right\}\right].$$

Using the approach similar to the one used to bound $(\blacksquare)_i$ in (6), we have

$$\mathbb{E}\left[\sum_{t=t_1}^{T} \mathbb{I}\left\{A_t = i \text{ and } \hat{\mu}_i(t-1) \geq \mu_1 + \frac{\Delta_i - c}{2}\right\}\right] \leq \frac{8}{(\Delta_i - c)^2}.$$

By applying Lemma 5, we can get

$$(\bigstar)_i \leq t_1(b, \mu, c) + \beta(b, \mu, K) + \frac{8}{(\Delta_i - c)^2}$$

where the term $\beta(b, \mu, K)$ is subsequently defined in Lemma 5.

Substituting the above inequality and (15) into (14), we arrive at

$$\mathbb{E}[N_i^{(0)}(T)] \leq 1 + \frac{1}{Kc} + t_1(b, \mu, c) + \beta(b, \mu, K) + \frac{8}{(\Delta_i - c)^2}$$
$$= o(\log T).$$

Due to the asymptotic optimality of Less-Exploring Thompson Sampling (Jin et al., 2023), for any suboptimal arm $i$, we have

$$\mathbb{E}[N_i(T)] \leq \frac{2 \log T}{\Delta_i^2} + o(\log T).$$

Finally, based on the regret decomposition in (3), we can conclude

$$R_{\mu,c}^{\mathrm{RG}}(T) = c \cdot \mathbb{E}[N_1^{(1)}(T)] + \sum_{i>1} \left( \Delta_i \cdot \mathbb{E}[N_i^{(0)}(T)] + c \cdot \mathbb{E}[N_i^{(1)}(T)] \right)$$
$$= c \cdot \mathbb{E}[N_1^{(1)}(T)] + \sum_{i>1} (\Delta_i \wedge c) \cdot \mathbb{E}[N_i(T)]$$
$$+ \sum_{i>1} \left( (\Delta_i - \Delta_i \wedge c) \cdot \mathbb{E}[N_i^{(0)}(T)] + (c - \Delta_i \wedge c) \cdot \mathbb{E}[N_i^{(1)}(T)] \right)$$
$$= \sum_{i>1} (\Delta_i \wedge c) \cdot \mathbb{E}[N_i(T)] + \sum_{i:\Delta_i < c} (c - \Delta_i) \cdot \mathbb{E}[N_i^{(1)}(T)] + \sum_{i:\Delta_i > c} (\Delta_i - c) \cdot \mathbb{E}[N_i^{(0)}(T)]$$
$$\leq (2 \log T) \sum_{i>1} \frac{\Delta_i \wedge c}{\Delta_i^2} + o(\log T)$$

where the second equality is due to the fact that $\mathbb{E}[N_i^{(0)}(T)] + \mathbb{E}[N_i^{(1)}(T)] = \mathbb{E}[N_i(T)]$ for all arms $i \in [K]$.

Therefore, it holds that

$$\limsup_{T \to \infty} \frac{R_{\mu,c}^{\mathrm{RG}}(T)}{\log T} \leq 2 \sum_{i>1} \frac{\Delta_i \wedge c}{\Delta_i^2}$$

as desired. $\qquad\square$

**Lemma 5.** *Consider Algorithm 1. For any $b \in (0, 1)$, there exists a constant $\beta(b, \mu, K)$ such that*

$$\sum_{t=1}^{\infty} \mathbb{P}\left( N_1(t) \leq t^b \right) \leq \beta(b, \mu, K).$$

*Proof of Lemma 5.* The proof of Lemma 5 is essentially the same as that of Proposition 5 in Korda et al. (2013), which was used to analyze the classical Thompson Sampling algorithm. In fact, the only difference between our arm sampling rule, which is built upon Less-Exploring Thompson Sampling (Jin et al., 2023), and the classical Thompson Sampling is how the estimated reward $a_i(t)$ is constructed for each arm $i \in [K]$. Specifically, in our arm sampling rule, $a_i(t)$ is either drawn from the posterior distribution $\mathcal{N}(\hat{\mu}_i(t-1), 1/N_i(t-1))$ with probability $1/K$ or set to be the empirical mean $\hat{\mu}_i(t-1)$ otherwise. In classical Thompson Sampling, $a_i(t)$ is always drawn from the posterior distribution. Therefore, it suffices to verify the parts concerning the probability distributions of $a_i(t)$; these correspond to Lemmas 9 and 10 in the proof of Proposition 5 in Korda et al. (2013).

It is straightforward to see that Lemma 9 in Korda et al. (2013) is applicable to our algorithm. For Lemma 10 therein, its counterpart is demonstrated in Lemma 6 below.

After establishing the counterparts of Lemmas 9 and 10 in the proof of Proposition 5 in Korda et al. (2013), we can extend the same analysis to our specific case. For the sake of completeness, we provide a proof sketch in the following.

Let $\tau_j$ denote the time of the $j$-th pull of the optimal arm (i.e., arm 1), with $\tau_0 := 0$. Define $\xi_j := (\tau_{j+1} - 1) - \tau_j$ as the random variable measuring the number of time steps between the

$j$-th and $(j+1)$-th pull of the optimal arm. With this setup, we can derive an upper bound for $\mathbb{P}\left(N_1(t) \leq t^b\right)$ as:

$$\mathbb{P}\left(N_1(t) \leq t^b\right) \leq \mathbb{P}\left(\exists j \in \left\{0,..,\left\lfloor t^b \right\rfloor\right\} : \xi_j \geq t^{1-b} - 1\right) \leq \sum_{j=0}^{\left\lfloor t^b \right\rfloor} \mathbb{P}(\xi_j \geq t^{1-b} - 1).$$

Consider the interval $\mathcal{I}_j := \left\{\tau_j, \ldots, \tau_j + \left\lceil t^{1-b} - 1 \right\rceil\right\}$. If $\xi_j \geq t^{1-b} - 1$, then no pull of the optimal arm occurs on $\mathcal{I}_j$.

The subsequent analysis aims to bound the probability that no pull of the optimal arm occurs within the interval $\mathcal{I}_j$. It relies on two key principles:

- First, for a suboptimal arm, if it has been pulled a sufficient number of times, then, with high probability, its estimated reward (sample) cannot deviate significantly from its true mean. This observation is quantitatively characterized in Lemma 10 of Korda et al. (2013), corresponding to Lemma 6 in our paper.

- Second, for the optimal arm, the probability that its estimated reward (sample) deviates significantly below its true mean during a long subinterval of $\mathcal{I}_j$ is low. This observation is quantitatively characterized in Lemma 9 of Korda et al. (2013), which directly applies to our case.

$\square$

**Lemma 6.** *Consider Algorithm 1. For all $t \in \mathbb{N}$, it holds that*

$$\mathbb{P}\left(\exists s \leq t, \exists i > 1 : a_i(s) > \mu_i + \frac{\Delta_i}{2}, N_i(s-1) > \frac{128 \log t}{\Delta_i^2}\right) \leq \frac{K}{t^2}.$$

*Proof of Lemma 6.* For any fixed $s \leq t$ and $i > 1$, we have

$$\begin{aligned}
&\mathbb{P}\left(a_i(s) > \mu_i + \frac{\Delta_i}{2}, N_i(s-1) > \frac{128 \log t}{\Delta_i^2}\right) \\
&\leq \mathbb{P}\left(\hat{\mu}_i(s-1) > \mu_i + \frac{\Delta_i}{4}, N_i(s-1) > \frac{128 \log t}{\Delta_i^2}\right) \\
&\quad + \mathbb{P}\left(a_i(s) > \mu_i + \frac{\Delta_i}{2}, N_i(s-1) > \frac{128 \log t}{\Delta_i^2}, \hat{\mu}_i(s-1) \leq \mu_i + \frac{\Delta_i}{4}\right).
\end{aligned}$$ (16)

For the first term in (16), by Lemma 3, we can bound it as

$$\begin{aligned}
&\mathbb{P}\left(\hat{\mu}_i(s-1) > \mu_i + \frac{\Delta_i}{4}, N_i(s-1) > \frac{128 \log t}{\Delta_i^2}\right) \\
&\leq \sum_{x=\left\lceil \frac{128 \log t}{\Delta_i^2} \right\rceil}^{t} \mathbb{P}\left(\hat{\mu}_{ix} > \mu_i + \frac{\Delta_i}{4}, N_i(s-1) = x\right) \\
&\leq \sum_{x=\left\lceil \frac{128 \log t}{\Delta_i^2} \right\rceil}^{t} \mathbb{P}\left(\hat{\mu}_{ix} > \mu_i + \frac{\Delta_i}{4}\right) \\
&\leq \sum_{x=\left\lceil \frac{128 \log t}{\Delta_i^2} \right\rceil}^{t} \frac{1}{t^4} \\
&\leq \frac{1}{t^3}.
\end{aligned}$$

For the second term in (16), according to the construction of $a_i(s)$ in Algorithm 1 and Lemma 3, we can get

$$\mathbb{P}\left(a_i(s) > \mu_i + \frac{\Delta_i}{2}, N_i(s-1) > \frac{128 \log t}{\Delta_i^2}, \hat{\mu}_i(s-1) \leq \mu_i + \frac{\Delta_i}{4}\right)$$

$$\leq \mathbb{P}\left(a_i(s) > \hat{\mu}_i(s-1) + \frac{\Delta_i}{4}, N_i(s-1) > \frac{128 \log t}{\Delta_i^2}\right)$$

$$\leq \sum_{x=\left\lceil \frac{128 \log t}{\Delta_i^2}\right\rceil}^{t} \mathbb{P}\left(a_i(s) > \hat{\mu}_i(s-1) + \frac{\Delta_i}{4}, N_i(s-1) = x\right)$$

$$\leq \sum_{x=\left\lceil \frac{128 \log t}{\Delta_i^2}\right\rceil}^{t} \frac{1}{K} \cdot \frac{1}{t^4}$$

$$\leq \frac{1}{Kt^3}.$$

Thus, for any $s \leq t$ and $i > 1$, it holds that

$$\mathbb{P}\left(a_i(s) > \mu_i + \frac{\Delta_i}{2}, N_i(s-1) > \frac{128 \log t}{\Delta_i^2}\right) \leq \frac{1}{t^3} + \frac{1}{Kt^3} = \frac{K+1}{Kt^3}.$$

Finally, by a union bound, we can conclude

$$\mathbb{P}\left(\exists s \leq t, \exists i > 1 : a_i(s) > \mu_i + \frac{\Delta_i}{2}, N_i(s-1) > \frac{128 \log t}{\Delta_i^2}\right) \leq \frac{(K+1)(K-1)}{Kt^2} \leq \frac{K}{t^2}.$$

$\square$

## C.2 LOWER BOUNDS

*Proof of Theorem 2.* In the following, we will establish the asymptotic and minimax lower bounds, respectively.

**Asymptotic lower bound.** Consider any algorithm $\pi$ that is $R^{\mathrm{RG}}$-consistent. Sine $\mathbb{E}[N_i^{(0)}(T)] + \mathbb{E}[N_i^{(1)}(T)] = \mathbb{E}[N_i(T)]$ for all arms $i \in [K]$, we can utilize the regret decomposition in (3) to derive

$$R_{\mu,c}^{\mathrm{RG}}(T,\pi) \geq \sum_{i>1}\left(\Delta_i \cdot \mathbb{E}[N_i^{(0)}(T)] + c \cdot \mathbb{E}[N_i^{(1)}(T)]\right)$$

$$\geq \sum_{i>1}\left((\Delta_i \wedge c) \cdot \mathbb{E}[N_i(T)]\right).$$

Fix any abstention regret $c > 0$. Then for all bandit instances $\mu \in \mathcal{U}$ and $a > 0$, it holds that

$$R_{\mu}^{\mathrm{CA}}(T,\pi) = T\mu_1 - \mathbb{E}\left[\sum_{t=1}^{T} X_t\right]$$

$$= \sum_{i>1}\left(\Delta_i \cdot \mathbb{E}[N_i(T)]\right)$$

$$\leq \max_{i>1} \frac{\Delta_i}{\Delta_i \wedge c} \cdot \sum_{i>1}\left((\Delta_i \wedge c) \cdot \mathbb{E}[N_i(T)]\right)$$

$$\leq \max_{i>1} \frac{\Delta_i}{\Delta_i \wedge c} \cdot R_{\mu,c}^{\mathrm{RG}}(T,\pi)$$

$$= o(T^a).$$

Therefore, in accordance with Definition 3, the algorithm $\pi$ is also $R^{\mathrm{CA}}$-consistent for arbitrary abstention regret $c$.

Subsequently, for any abstention regret $c$ and bandit instance $\mu$, we have

$$\liminf_{T\to\infty} \frac{R^{\mathrm{RG}}_{\mu,c}(T,\pi)}{\log T} \geq \liminf_{T\to\infty} \frac{\sum_{i>1}\left((\Delta_i \wedge c)\cdot \mathbb{E}[N_i(T)]\right)}{\log T}$$
$$= \sum_{i>1}(\Delta_i \wedge c)\cdot \liminf_{T\to\infty}\frac{\mathbb{E}[N_i(T)]}{\log T}$$
$$\geq 2\sum_{i>1}\frac{\Delta_i \wedge c}{\Delta_i^2},$$

where the last inequality follows from the property of $R^{\mathrm{CA}}$-consistent policies as detailed in Remark 7.

This concludes the proof of the instance-dependent asymptotic lower bound.

**Minimax lower bound.** We extend the proof of the minimax lower bound from the canonical multi-armed bandit model to the model incorporating fixed-regret abstention.

Consider any fixed abstention regret $c > 0$, time horizon $T \geq K$ and algorithm $\pi \in \Pi^{\mathrm{RG}}$. We construct a bandit instance $\mu \in \mathcal{U}$, where $\mu_1 = \Delta$ and $\mu_i = 0$ for all $i \in [K]\setminus\{1\}$. Here, $\Delta > 0$ is some parameter whose exact value will be determined later. We use $\mathbb{P}_{\mu,c}$ to represent the probability distribution of the sequence $(A_1, B_1, X_1, \ldots, A_T, B_T, X_T)$ induced by the algorithm $\pi$ for the abstention regret $c$ and bandit instance $\mu$. Since $\sum_{i=1}\mathbb{E}_{\mu,c}[N_i(T)] = T$, according to the pigeonhole principle, there must exist an index $j \in [K]\setminus\{1\}$ such that

$$\mathbb{E}_{\mu,c}[N_j(T)] \leq \frac{T}{K-1}.$$

Now we construct another bandit instance $\mu' \in \mathcal{U}$, where $\mu'_1 = \Delta$, $\mu'_j = 2\Delta$ and $\mu'_i = 0$ for all $i \in [K]\setminus\{1,j\}$. Let $\mathbb{P}_{\mu',c}$ denote the probability distribution of the sequence $(A_1, B_1, X_1, \ldots, A_T, B_T, X_T)$ induced by the algorithm $\pi$ for the abstention regret $c$ and bandit instance $\mu'$.

For the first bandit instance $\mu$, regardless of the abstention option, if $N_1(T) \leq T/2$, then the cumulative regret must be at least $(\Delta \wedge c)T/2$. Therefore, we have

$$R^{\mathrm{RG}}_{\mu,c}(T,\pi) \geq \frac{(\Delta \wedge c)T}{2}\mathbb{P}_{\mu,c}(N_1(T) \leq T/2).$$

Similarly, for the second bandit instance $\mu'$, we can obtain

$$R^{\mathrm{RG}}_{\mu',c}(T,\pi) \geq \frac{(\Delta \wedge c)T}{2}\mathbb{P}_{\mu',c}(N_1(T) > T/2).$$

By combining the aforementioned two inequalities and applying Lemma 1, we obtain the following:

$$R^{\mathrm{RG}}_{\mu,c}(T,\pi) + R^{\mathrm{RG}}_{\mu',c}(T,\pi) \geq \frac{(\Delta \wedge c)T}{2}\left(\mathbb{P}_{\mu,c}(N_1(T) \leq T/2) + \mathbb{P}_{\mu',c}(N_1(T) > T/2)\right)$$
$$\geq \frac{(\Delta \wedge c)T}{4}\exp\left(-\mathrm{KL}\left(\mathbb{P}_{\mu,c}, \mathbb{P}_{\mu',c}\right)\right).$$

Leveraging Lemma 2 and the KL divergence between Gaussian distributions, we can derive

$$\mathrm{KL}\left(\mathbb{P}_{\mu,c}, \mathbb{P}_{\mu',c}\right) = \mathbb{E}_{\mu,c}[N_j(T)]\frac{(2\Delta)^2}{2} \leq \frac{2T\Delta^2}{K-1}.$$

Altogether, we can arrive at

$$R^{\mathrm{RG}}_{\mu,c}(T,\pi) + R^{\mathrm{RG}}_{\mu',c}(T,\pi) \geq \frac{(\Delta \wedge c)T}{4}\exp\left(-\frac{2T\Delta^2}{K-1}\right).$$

Now, we set $\Delta = \sqrt{\frac{K}{T}} \wedge c$, which leads to

$$R^{\mathrm{RG}}_{\mu,c}(T,\pi) + R^{\mathrm{RG}}_{\mu',c}(T,\pi) \geq \frac{\sqrt{KT} \wedge cT}{4} \exp\left(-\frac{2T(\sqrt{\frac{K}{T}} \wedge c)^2}{K-1}\right)$$

$$\geq \frac{\sqrt{KT} \wedge cT}{4} \exp\left(-\frac{2K}{K-1}\right)$$

$$\geq \frac{\exp(-4)}{4}(\sqrt{KT} \wedge cT).$$

Consequently, either $R^{\mathrm{RG}}_{\mu,c}(T,\pi)$ or $R^{\mathrm{RG}}_{\mu',c}(T,\pi)$ is at least $\frac{\exp(-4)}{8}(\sqrt{KT} \wedge cT)$, which completes the proof of the instance-independent minimax lower bound. □

# D ANALYSIS OF THE FIXED-REWARD SETTING

## D.1 UPPER BOUNDS

*Proof of Theorem 3.* Utilizing the law of total expectation, we can decompose the regret $R^{\mathrm{RW}}_{\mu,c}(T,\pi)$ in the following:

$$R^{\mathrm{RW}}_{\mu,c}(T,\pi) = T \cdot (\mu_1 \vee c) - \mathbb{E}\left[\sum_{t=1}^{T}\left(X_t \cdot \mathbb{1}\{B_t = 0\} + c \cdot \mathbb{1}\{B_t = 1\}\right)\right]$$

$$= T \cdot (\mu_1 \vee c) - \mathbb{E}\left[\sum_{t=1}^{T}\left(\mu_{A_t} \cdot \mathbb{1}\{B_t = 0\} + c \cdot \mathbb{1}\{B_t = 1\}\right)\right]$$

$$= \sum_{i \in [K]}\left((\mu_1 \vee c - \mu_i) \cdot \mathbb{E}[N_i^{(0)}(T)] + (\mu_1 \vee c - c) \cdot \mathbb{E}[N_i^{(1)}(T)]\right). \qquad (17)$$

Recall that we define $\hat{\mu}_i(t) = +\infty$ if $N_i(t) = 0$ for all arms $i \in [K]$. Thus, for any arm $i$ with $\mu_i > c$, we can obtain

$$\mathbb{E}[N_i^{(1)}(T)] = \mathbb{E}\left[\sum_{t=1}^{T}\mathbb{I}\{A_t = i \text{ and } B_t = 1\}\right]$$

$$= \mathbb{E}\left[\sum_{t=1}^{T}\mathbb{I}\{A_t = i \text{ and } \hat{\mu}_i(t-1) \leq c\}\right]$$

$$\leq \mathbb{E}\left[\sum_{t=1}^{T}\sum_{s=0}^{T-1}\mathbb{I}\{A_t = i \text{ and } \hat{\mu}_{is} \leq c \text{ and } N_i(t-1) = s\}\right]$$

$$= \mathbb{E}\left[\sum_{t=1}^{T}\sum_{s=1}^{T-1}\mathbb{I}\{A_t = i \text{ and } \hat{\mu}_{is} \leq c \text{ and } N_i(t-1) = s\}\right]$$

$$\leq \mathbb{E}\left[\sum_{s=1}^{T-1}\mathbb{I}\{\hat{\mu}_{is} \leq c\}\right]$$

$$\leq \frac{2}{(\mu_i - c)^2}$$

where the penultimate inequality is derived from an argument analogous to that in (5), and the last inequality is a consequence of Lemma 4.

Similarly, for any arm $i$ with $\mu_i < c$, we have

$$
\begin{aligned}
\mathbb{E}[N_i^{(0)}(T)] &= \mathbb{E}\left[\sum_{t=1}^{T} \mathbb{I}\{A_t = i \text{ and } B_t = 0\}\right] \\
&= \mathbb{E}\left[\sum_{t=1}^{T} \mathbb{I}\{A_t = i \text{ and } \hat{\mu}_i(t-1) > c\}\right] \\
&\leq \mathbb{E}\left[\sum_{t=1}^{T}\sum_{s=0}^{T-1} \mathbb{I}\{A_t = i \text{ and } \hat{\mu}_{is} > c \text{ and } N_i(t-1) = s\}\right] \\
&\leq 1 + \mathbb{E}\left[\sum_{t=1}^{T}\sum_{s=1}^{T-1} \mathbb{I}\{A_t = i \text{ and } \hat{\mu}_{is} > c \text{ and } N_i(t-1) = s\}\right] \\
&\leq 1 + \frac{2}{(c - \mu_i)^2}.
\end{aligned}
$$

**Asymptotic upper bound.** First, we consider the scenario where $\mu_1 \leq c$. In this case, based on the regret decomposition in Equation (17), we can bound the regret as follows:

$$
\begin{aligned}
R_{\mu,c}^{\mathrm{RW}}(T) &= \sum_{i \in [K]} (c - \mu_i) \cdot \mathbb{E}[N_i^{(0)}(T)] \\
&= \sum_{i:\mu_i < c} (c - \mu_i) \cdot \mathbb{E}[N_i^{(0)}(T)] \\
&\leq \sum_{i:\mu_i < c} \left(c - \mu_i + \frac{2}{c - \mu_i}\right) \\
&= o(\log T).
\end{aligned}
\tag{18}
$$

Next, we consider the scenario where $\mu_1 > c$. Due to the asymptotic optimality of the base algorithm, for any suboptimal arm $i$, we have

$$
\mathbb{E}[N_i(T)] \leq \frac{2 \log T}{\Delta_i^2} + o(\log T).
$$

Thus, we can bound the regret as:

$$
\begin{aligned}
&R_{\mu,c}^{\mathrm{RW}}(T) \\
&= \sum_{i \in [K]} \left((\mu_1 - \mu_i) \cdot \mathbb{E}[N_i^{(0)}(T)] + (\mu_1 - c) \cdot \mathbb{E}[N_i^{(1)}(T)]\right) \\
&= \sum_{i \in [K]} \left((\mu_1 - \mu_i \vee c + \mu_i \vee c - \mu_i) \cdot \mathbb{E}[N_i^{(0)}(T)] + (\mu_1 - \mu_i \vee c + \mu_i \vee c - c) \cdot \mathbb{E}[N_i^{(1)}(T)]\right) \\
&= \sum_{i \in [K]} \left((\mu_1 - \mu_i \vee c) \cdot \mathbb{E}[N_i(T)] + (\mu_i \vee c - \mu_i) \cdot \mathbb{E}[N_i^{(0)}(T)] + (\mu_i \vee c - c) \cdot \mathbb{E}[N_i^{(1)}(T)]\right) \\
&= \sum_{i > 1} (\mu_1 - \mu_i \vee c) \cdot \mathbb{E}[N_i(T)] + \sum_{i:\mu_i < c} (c - \mu_i) \cdot \mathbb{E}[N_i^{(0}(T)] + \sum_{i:\mu_i > c} (\mu_i - c) \cdot \mathbb{E}[N_i^{(1)}(T)] \\
&\leq (2 \log T) \sum_{i > 1} \frac{\mu_1 - \mu_i \vee c}{\Delta_i^2} + \sum_{i:\mu_i < c} \left(c - \mu_i + \frac{2}{c - \mu_i}\right) + \sum_{i:\mu_i > c} \frac{2}{\mu_i - c} + o(\log T) \\
&= (2 \log T) \sum_{i > 1} \frac{\mu_1 - \mu_i \vee c}{\Delta_i^2} + o(\log T)
\end{aligned}
$$

where the third equality is due to the fact that $\mathbb{E}[N_i^{(0)}(T)] + \mathbb{E}[N_i^{(1)}(T)] = \mathbb{E}[N_i(T)]$ for all arms $i \in [K]$.

Altogether, in both scenarios, it holds that

$$
\limsup_{T \to \infty} \frac{R_{\mu,c}^{\mathrm{RW}}(T)}{\log T} \leq 2 \sum_{i > 1} \frac{\mu_1 \vee c - \mu_i \vee c}{\Delta_i^2}.
$$

**Minimax upper bound.** First, if $\mu_1 \leq c$, by utilizing the regret decomposition in Equation (18), we have

$$
\begin{aligned}
R_{\mu,c}^{\mathrm{RW}}(T) &= \sum_{i:\mu_i < c} (c - \mu_i) \cdot \mathbb{E}[N_i^{(0)}(T)] \\
&= \sum_{i:0 < c-\mu_i < \sqrt{\frac{K}{T}}} (c - \mu_i) \cdot \mathbb{E}[N_i^{(0)}(T)] + \sum_{i:c-\mu_i \geq \sqrt{\frac{K}{T}}} (c - \mu_i) \cdot \mathbb{E}[N_i^{(0)}(T)] \\
&\leq \sqrt{\frac{K}{T}} \sum_{i:0 < c-\mu_i < \sqrt{\frac{K}{T}}} \mathbb{E}[N_i^{(0)}(T)] + \sum_{i:c-\mu_i \geq \sqrt{\frac{K}{T}}} \left( c - \mu_i + \frac{2}{c - \mu_i} \right) \\
&\leq \sqrt{\frac{K}{T}} \cdot T + \sum_{i:c-\mu_i \geq \sqrt{\frac{K}{T}}} \left( c - \mu_i + 2\sqrt{\frac{T}{K}} \right) \\
&\leq \sqrt{KT} + \sum_{i \in [K]} (c - \mu_i) + K \cdot 2\sqrt{\frac{T}{K}} \\
&= 3\sqrt{KT} + \sum_{i \in [K]} (\mu_1 \vee c - \mu_i).
\end{aligned}
\tag{19}
$$

Next, if $\mu_1 > c$, then the best possible (expected) reward at a single time step is $\mu_1$, which coincides with the canonical multi-armed bandit problem. Consequently, compared with canonical multi-armed bandits, at a single time step, the agent in our problem incurs a greater (expected) regret only if an arm $i$ with $\mu_i > c$ is pulled and the abstention option is chosen. Thus, we have

$$
R_{\mu,c}^{\mathrm{RW}}(T) \leq R_{\mu}^{\mathrm{CA}}(T) + \sum_{i:\mu_i > c} (\mu_i - c) \cdot \mathbb{E}[N_i^{(1)}(T)].
$$

Due to the minimax optimality of the base algorithm, there exists a universal constant $\alpha_1 > 0$ such that

$$
R_{\mu}^{\mathrm{CA}}(T) \leq \alpha_1 \left( \sqrt{KT} + \sum_{i > 1} \Delta_i \right).
$$

Furthermore, using a similar argument as in (19), we can derive

$$
\begin{aligned}
&\sum_{i:\mu_i > c} (\mu_i - c) \cdot \mathbb{E}[N_i^{(1)}(T)] \\
&= \sum_{i:0 < \mu_i - c < \sqrt{\frac{K}{T}}} (\mu_i - c) \cdot \mathbb{E}[N_i^{(1)}(T)] + \sum_{i:\mu_i - c \geq \sqrt{\frac{K}{T}}} (\mu_i - c) \cdot \mathbb{E}[N_i^{(1)}(T)] \\
&\leq \sqrt{KT} + \sum_{i:\mu_i - c \geq \sqrt{\frac{K}{T}}} \frac{2}{\mu_i - c} \\
&\leq 3\sqrt{KT}.
\end{aligned}
$$

Therefore, we can bound the regret as

$$
\begin{aligned}
R_{\mu,c}^{\mathrm{RW}}(T) &\leq (\alpha_1 + 3) \left( \sqrt{KT} + \sum_{i > 1} \Delta_i \right) \\
&= (\alpha_1 + 3) \left( \sqrt{KT} + \sum_{i \in [K]} (\mu_1 \vee c - \mu_i) \right).
\end{aligned}
$$

As a result, the desired minimax upper bound holds in both scenarios. □

### D.2 LOWER BOUNDS

*Proof of Theorem 4.* The proof structure for Theorem 4 closely parallels that of Theorem 2 in Appendix C.2, although certain specific details contain significant variations. Therefore, we will streamline the shared components and elaborate on the distinctions.

**Asymptotic lower bound.** Consider any $R^{\text{RW}}$-consistent algorithm $\pi$ and bandit instance $\mu \in \mathcal{U}$. The case that $c \geq \mu_1$ is trivial, as $R_{\mu,c}^{\text{RW}}(T, \pi)$ is non-negative, and $\mu_1 \vee c - \mu_i \vee c = 0$ for all $i > 1$. Thus, it suffices to demonstrate that for any abstention reward $c < \mu_1$,

$$\liminf_{T \to \infty} \frac{R_{\mu,c}^{\text{RW}}(T, \pi)}{\log T} \geq 2 \sum_{i>1} \frac{\mu_1 - \mu_i \vee c}{\Delta_i^2}.$$

When $c < \mu_1$, we can establish a lower bound on $R_{\mu,c}^{\text{RW}}(T, \pi)$ as follows:

$$
\begin{aligned}
R_{\mu,c}^{\text{RW}}(T, \pi) &= \sum_{i \in [K]} \left( (\mu_1 - \mu_i) \cdot \mathbb{E}[N_i^{(0)}(T)] + (\mu_1 - c) \cdot \mathbb{E}[N_i^{(1)}(T)] \right) \\
&\geq \sum_{i>1} \left( (\mu_1 - \mu_i) \cdot \mathbb{E}[N_i^{(0)}(T)] + (\mu_1 - c) \cdot \mathbb{E}[N_i^{(1)}(T)] \right) \\
&\geq \sum_{i>1} \left( (\mu_1 - \mu_i \vee c) \cdot \mathbb{E}[N_i(T)] \right).
\end{aligned}
$$

Therefore, we only need to show for all suboptimal arms $i > 1$,

$$\liminf_{T \to \infty} \frac{\mathbb{E}[N_i(T)]}{\log T} \geq \frac{2}{\Delta_i^2}. \tag{20}$$

However, unlike the asymptotic lower bound part of the proof of Theorem 2, we cannot apply the properties of $R^{\text{CA}}$-consistency here, as $R^{\text{RW}}$-consistency does not imply $R^{\text{CA}}$-consistency in general. Instead, we will demonstrate the desired result (20) directly.

Fix an index $j > 1$ and take $\varepsilon > 0$. We now proceed to create an alternative bandit instance $\mu' \in \mathcal{U}$, where $\mu_j' = \mu_1 + \varepsilon$ and $\mu_i' = \mu_i$ for all $i \in [K] \setminus \{j\}$. Note that for the new bandit instance, it holds that $\max_{i \in [K]} \mu_i' = \mu_j' > \mu_1 > c$. To distinguish between the two scenarios, we will refer to the probability distribution associated with the sequence $(A_1, B_1, X_1, \ldots, A_T, B_T, X_T)$, generated by the algorithm $\pi$ for the abstention reward $c$ and the original bandit scenario $\mu$, as $\mathbb{P}_{\mu,c}$, and for the new bandit instance $\mu'$, we denote the corresponding distribution as $\mathbb{P}_{\mu',c}$.

A straightforward computation yields

$$R_{\mu,c}^{\text{RW}}(T, \pi) \geq \frac{(\mu_1 - \mu_i \vee c)T}{2} \mathbb{P}_{\mu,c}(N_j(T) > T/2)$$

and

$$R_{\mu',c}^{\text{RW}}(T, \pi) \geq \frac{\varepsilon T}{2} \mathbb{P}_{\mu',c}(N_j(T) \leq T/2).$$

Employing a similar approach to the one used in the minimax lower bound part of the proof of Theorem 2, utilizing Lemmas 1 and 2, we can derive:

$$
\begin{aligned}
R_{\mu,c}^{\text{RW}}(T, \pi) + R_{\mu',c}^{\text{RW}}(T, \pi) &\geq \frac{((\mu_1 - \mu_i \vee c) \wedge \varepsilon)T}{2} \left( \mathbb{P}_{\mu,c}(N_j(T) > T/2) + \mathbb{P}_{\mu',c}(N_j(T) \leq T/2) \right) \\
&\geq \frac{((\mu_1 - \mu_i \vee c) \wedge \varepsilon)T}{4} \exp\left( -\text{KL}\left( \mathbb{P}_{\mu,c}, \mathbb{P}_{\mu',c} \right) \right) \\
&= \frac{((\mu_1 - \mu_i \vee c) \wedge \varepsilon)T}{4} \exp\left( -\mathbb{E}_{\mu,c}[N_j(T)] \frac{(\Delta_j + \varepsilon)^2}{2} \right).
\end{aligned}
$$

By rearranging the above inequality and taking the limit inferior, we have

$$\liminf_{T \to \infty} \frac{\mathbb{E}[N_j(T)]}{\log T} \geq \frac{2}{(\Delta_j + \varepsilon)^2} \left( 1 + \limsup_{T \to \infty} \frac{\log\left( \frac{(\mu_1 - \mu_i \vee c) \wedge \varepsilon}{4\left(R_{\mu,c}^{\mathrm{RW}}(T,\pi) + R_{\mu',c}^{\mathrm{RW}}(T,\pi)\right)} \right)}{\log T} \right)$$

$$= \frac{2}{(\Delta_j + \varepsilon)^2} \left( 1 - \limsup_{T \to \infty} \frac{\log\left( R_{\mu,c}^{\mathrm{RW}}(T,\pi) + R_{\mu',c}^{\mathrm{RW}}(T,\pi) \right)}{\log T} \right).$$

Recall the definition of $R^{\mathrm{RW}}$-consistency. For all $a > 0$, both $R_{\mu,c}^{\mathrm{RW}}(T,\pi)$ and $R_{\mu',c}^{\mathrm{RW}}(T,\pi)$ are on the order of $o(T^a)$, and hence,

$$\limsup_{T \to \infty} \frac{\log\left( R_{\mu,c}^{\mathrm{RW}}(T,\pi) + R_{\mu',c}^{\mathrm{RW}}(T,\pi) \right)}{\log T} \leq a.$$

By letting both $a$ and $\varepsilon$ approach zero, we can establish the desired result (20), thereby concluding the proof of the asymptotic lower bound.

**Minimax lower bound.** The construction employed in the fixed-reward setting here is analogous to the one utilized in the proof of Theorem 2.

Consider any fixed abstention reward $c \in \mathbb{R}$, time horizon $T \geq K$ and algorithm $\pi \in \Pi^{\mathrm{RW}}$. Let $\Delta > 0$ be a parameter to be determined later. We construct a bandit instance $\mu \in \mathcal{U}$, where $\mu_1 = \Delta + c$ and $\mu_i = c$ for all $i \in [K] \setminus \{1\}$. Note that there must exist an index $j \in [K] \setminus \{1\}$ such that $\mathbb{E}_{\mu,c}[N_j(T)] \leq \frac{T}{K-1}$. We then construct another bandit instance $\mu' \in \mathcal{U}$, where $\mu_1' = \Delta + c$, $\mu_j' = 2\Delta + c$ and $\mu_i' = c$ for all $i \in [K] \setminus \{1, j\}$.

Similarly, by applying Lemmas 1 and 2, we can derive that

$$R_{\mu,c}^{\mathrm{RW}}(T,\pi) + R_{\mu',c}^{\mathrm{RW}}(T,\pi) \geq \frac{\Delta T}{2} \left( \mathbb{P}_{\mu,c}(N_1(T) \leq T/2) + \mathbb{P}_{\mu',c}(N_1(T) > T/2) \right)$$

$$\geq \frac{\Delta T}{4} \exp\left( -\mathrm{KL}\left( \mathbb{P}_{\mu,c}, \mathbb{P}_{\mu',c} \right) \right)$$

$$= \frac{\Delta T}{4} \exp\left( -\mathbb{E}_{\mu,c}[N_j(T)] \frac{(2\Delta)^2}{2} \right)$$

$$\geq \frac{\Delta T}{4} \exp\left( -\frac{2T\Delta^2}{K-1} \right).$$

By choosing $\Delta = \sqrt{\frac{K}{T}}$, we have

$$R_{\mu,c}^{\mathrm{RW}}(T,\pi) + R_{\mu',c}^{\mathrm{RW}}(T,\pi) \geq \frac{\exp(-4)}{8} \sqrt{KT}.$$

Consequently, either $R_{\mu,c}^{\mathrm{RW}}(T,\pi)$ or $R_{\mu',c}^{\mathrm{RW}}(T,\pi)$ is at least $\frac{\exp(-4)}{8} \sqrt{KT}$.

Therefore, we have established the instance-independent minimax lower bound.

$\square$

# E ADDITIONAL NUMERICAL RESULTS

## E.1 RESULTS FOR THE FIXED-REWARD SETTING

In this part, we present the empirical results pertaining to the fixed-reward setting. Specifically, we examine the empirical performances of two particular realizations of our algorithm FRW-ALGwA (as outlined in Algorithm 2): FRW-TSwA and FRW-UCBwA. The former uses Less-Exploring

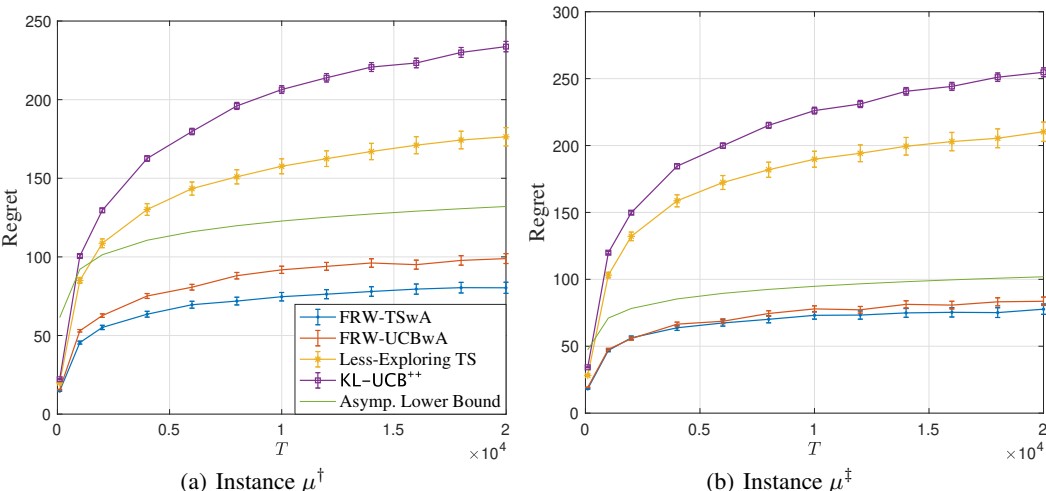

(a) Instance $\mu^\dagger$          (b) Instance $\mu^\ddagger$

Figure 4: Empirical regrets with abstention reward $c = 0.9$ for different time horizons $T$.

Thompson Sampling (Jin et al., 2023) as its base algorithm, while the latter employs KL-UCB$^{++}$ (Ménard & Garivier, 2017). Note that KL-UCB$^{++}$ is not an anytime algorithm; that is, it requires the prior knowledge of the time horizon $T$ as an input parameter. In the following experiments, we continue to utilize the two bandit instances $\mu^\dagger$ and $\mu^\ddagger$, as previously defined in Section 5.

In a manner analogous to our methodology for the fixed-regret setting, we adopt the original versions of Less-Exploring TS and KL-UCB$^{++}$ as baseline algorithms without the abstention option. The experimental results of the different methods with abstention reward $c = 0.9$ for different time horizons $T$ are presented in Figure 4. Additionally, we plot the instance-dependent asymptotic lower bound (ignoring the limit in $T$) on the cumulative regret (see Theorem 4) within each sub-figure. From Figure 4, we have the following observations:

- Both realizations of our algorithm, FRW-TSwA and FRW-UCBwA, exhibit marked superiority over the two non-abstaining baselines.

- Concerning the observed growth trend, as the time horizon $T$ increases, the performance curves for both FRW-TSwA and FRW-UCBwA approximate the asymptotic lower bound closely. This behavior indicates that the expected cumulative regrets of FRW-TSwA and FRW-UCBwA attain the instance-dependent lower bound asymptotically, validating the theoretical findings discussed in Section 4.

- While both FRW-TSwA and FRW-UCBwA represent implementations of our general algorithm and share identical theoretical guarantees, FRW-TSwA demonstrates superior empirical performance. This is particularly evident in the first instance $\mu^\dagger$, suggesting its enhanced applicability for real-world applications.

Next, we examine the impact of the abstention reward $c$ by assessing the performance of FRW-TSwA and FRW-UCBwA for different $c$, while keeping the time horizon $T$ fixed at $10,000$. The experimental results for bandit instances $\mu^\dagger$ and $\mu^\ddagger$ are shown in Figure 5.

Within each sub-figure, a pattern emerges. As the abstention reward $c$ increases, the empirical average cumulative regret initially remains relatively stable and starts to decline once $c$ crosses a certain threshold, eventually stabilizing around a small value. These observations are consistent with our theoretical expectations. Specifically, when the abstention reward $c$ is lower than the smallest mean reward among the arms, the agent derives no benefit from opting for the abstention action over selecting an arm. On the other hand, when the abstention reward $c$ exceeds the highest mean reward of the arms, abstention becomes the optimal decision and its reward is even superior to choosing the best arm. In this specific scenario, it is possible to achieve a regret of $o(\log T)$; see Remark 5 for further insights.

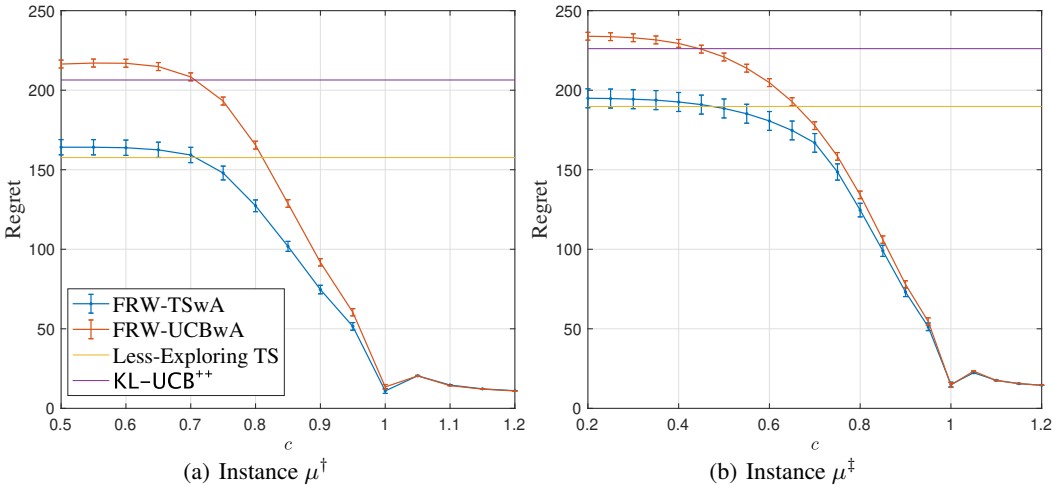

(a) Instance $\mu^\dagger$        (b) Instance $\mu^\ddagger$

Figure 5: Empirical regrets with time horizon $T = 10,000$ for different abstention rewards $c$.

### E.2 RANDOM INSTANCES

In this subappendix, we present additional numerical experiments, using random instances with large numbers of arms. The construction of these random instances mirrors the method in Jin et al. (2023). Specifically, for a given number of arms denoted by $K \geq 10$, we set $\mu_1 = 1$ and $\mu_i = 0.7$ for $i \in \{2, 3, \ldots, 10\}$, while $\mu_i \sim \text{Unif}[0.3, 0.5]$ for $i \in [K] \setminus [10]$.

For the sake of simplicity in presentation, we focus on the fixed-regret setting, examining two choices of $K$, namely, $K = 20$ and $K = 30$. The empirical averaged cumulative regrets with abstention regret $c = 0.1$ for different time horizons $T$ are shown in Figure 6, while the experimental results with time horizon $T = 10,000$ for different abstention rewards $c$ are illustrated in Figure 7.

It is evident that the findings in Figures 6 and 7 closely resemble those in Figures 2 and 3. Notably, FRW-TSwA outperforms Less-Exploring TS which is not tailored to the setting with the abstention option.

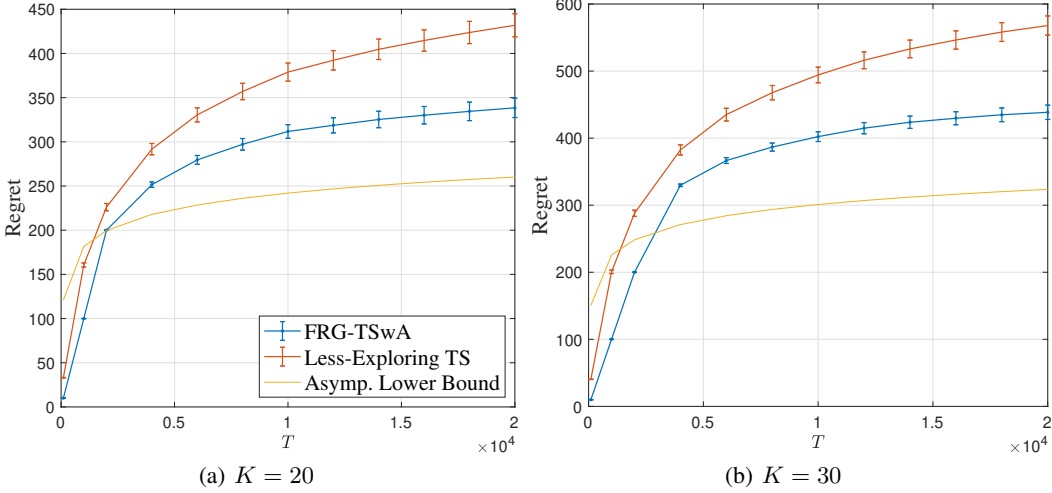

(a) $K = 20$        (b) $K = 30$

Figure 6: Empirical regrets with abstention regret $c = 0.1$ for different time horizons $T$.

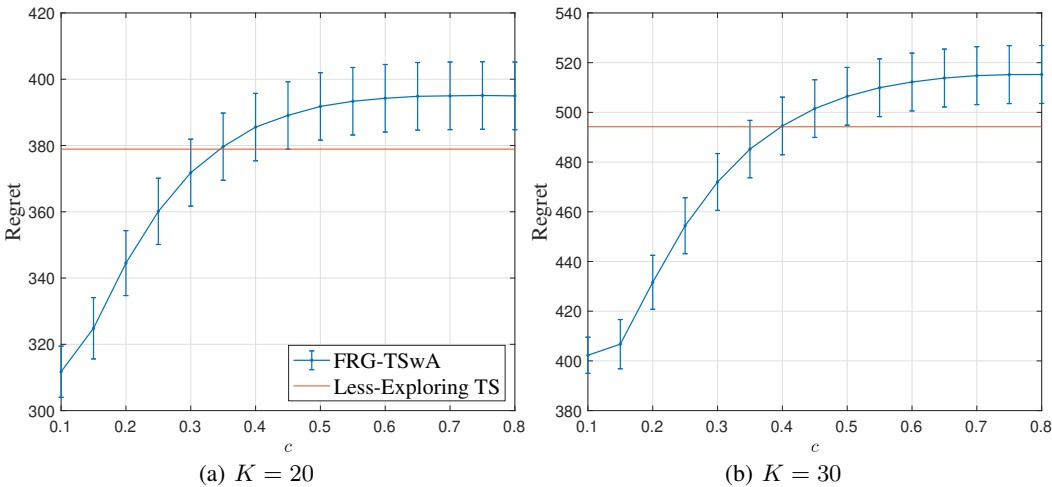

(a) $K = 20$ (b) $K = 30$

Figure 7: Empirical regrets with time horizon $T = 10,000$ for different abstention regrets $c$.

