# OpenReview forum: "Multi-Armed Bandits with Abstention"
_ICLR.cc/2024/Conference — Submitted to ICLR 2024_

### Official Review · Reviewer_Yyub · 2023-10-23

**Soundness:** 3 good
**Presentation:** 4 excellent
**Contribution:** 3 good
**Rating:** 8
**Confidence:** 3

**Summary:**

This paper studies a multi-armed bandit setting which features abstention, meaning that after selecting an arm, the agent has the option to suffer a fixed regret c (called fixed-regret setting) or to gain a fixed reward c (called fixed-reward setting). For each abstention model, authors propose lower bounds, an algorithm tackling the corresponding regret minimization problem, and its associated upper bounds, both in terms of asymptotic and minimax bounds.

**Strengths:**

- Originality : Algorithm 2, which allows to turn any “canonical” bandit algorithm into an abstention-featuring algorithm, is original while simple to implement.
- Quality : I did not check the proofs in Appendix in detail, but the results seem consistent (and good) and the outlines of the proof logical. The baselines and the different experimental settings considered in the experimental study and the Appendix make sense and are convincing.
- Clarity : I think the paper is very well-written, and distinguishes clearly the different settings, and types of theoretical results. The remarks are relevant and interesting.
- Significance : The topic of assessing whether adding abstention to an online learning allows to improve the performance of the agent is theoretically interesting.

**Weaknesses:**

- Significance : In terms of real-life applications (clinical trials), I am not totally convinced by the considered abstention model(s). In both fixed-regret and fixed-reward settings, the fact that the agent gets access to the stochastic reward no matter the value of Bt does not seem realistic, as it implies that there is some estimation of the reward/regret from pulling a treatment arm (referring to paragraph 3 on page 1). Please correct me if I am wrong.

**Questions:**

- In the experimental sections (both in the main paper for the fixed-regret setting and in the Appendix for the fixed-reward setting), in the plots at fixed T=10,000, perhaps it would be fairer to report the lines corresponding to the regret incurred by the baselines at T=10,000 in each setting, as the baselines seem to fare (slightly) better than the proposed algorithms for lower values of c (in the fixed-reward setting: ex. c=0.5 in the first instance) and for greater values of c (in the fixed-regret setting: ex: c=0.4 in the first instance) than shown in the empirical regret curves.
- epsilon(=1/K)-TS is asymptotically and minimax optimal for other one-dimensional exponential families (Poisson, Gamma, etc.). Since your algorithmic contributions partially rely on the analysis of epsilon-TS, how hard would it be to extend your results beyond Gaussian distributions?

**Details Of Ethics Concerns:**

None.

---

> ### Author Response · Authors · 2023-11-14
>
> We express our gratitude to the reviewer for the favorable assessment of our paper and for providing us with constructive feedback. We would like to respond to the questions in the following.
>
> ---
>
> **Weakness (Significance):**
>
> We do not fully understand concerns raised by the reviewer, and we welcome any additional comments. In particular, we do not fully understand the phrase "no matter the value of $B_t$ does not seem realistic, as it implies that there is some estimation of the reward/regret from pulling a treatment arm". The agent indeed observes the stochastic outcome $X_t$ after selecting an arm $A_t \in [K] $ and the abstention option $B_t\in \\{0,1\\}$.   In the example of clinical trials presented in the introduction, if researchers implement costly prearranged measures, the treatment outcomes will not contribute to the optimization objective of the study. However, the researcher can still observe these outcomes, thereby leading to more efficient future experimental designs. If the researcher is unable to observe the outcomes, then conducting medical treatments becomes meaningless. In line with this model assumption, our model well aligns with Neu \& Zhivotovskiy (2020), which explored the role of abstention in the context of online prediction with expert advice.
>
> The abstention decision $B_t$ must be made before observing the outcome $X_t$. Consequently in each time step, the agent utilizes  estimates of the reward/regret from pulling the chosen arm to make the abstention decision.
>
> > Gergely Neu and Nikita Zhivotovskiy. Fast rates for online prediction with abstention. In Conference on Learning Theory, pp. 3030–3048. PMLR, 2020.
>
> ---
>
> **Question 1 (Experiments):**
>
>
> We extend our sincere appreciation to the reviewer for offering constructive suggestions. In the revised manuscript, we have presented the performance of baselines in Figures 3 and 5, and provided a concise summary of the subsequent explanation. When the abstention reward $c$ is lower than the smallest mean reward in the fixed-reward setting, or the abstention regret $c$ exceeds the largest suboptimality gap in the fixed-regret setting, the agent gains no advantage in choosing the abstention option when selecting any arm. Unfortunately, the agent lacks precise knowledge of the bandit instance, leading to the inevitable selection of the abstention option. This unavoidable choice introduces a subtle gap between the performance of our algorithms and the baselines.
>
> ---
>
> **Question 2 (Extension beyond Gaussian distributions):**
>
> Epsilon(=1/K)-TS is asymptotically and minimax optimal for various popular reward distributions, including Gaussian, Bernoulli, Poisson, and Gamma. The extension beyond Gaussian distributions appears to be straightforward based on our current analysis, although careful consideration is necessary for specific "boundary" cases. For instance, in Bernoulli bandits where the stochastic rewards are bounded in $[0, 1]$, the behavior of minimax regrets might be different from the canonical case of $O(\sqrt{KT})$, when the abstention reward approaches 1 in the fixed-reward setting.
>
> ---
>
> Thank you once more for providing us with your valuable feedback. We sincerely hope that our responses effectively address the raised concerns.

---

> > ### Comment · Reviewer_Yyub · 2023-11-15
> > **Response to the rebuttal**
> >
> > Thanks to the authors for addressing my concerns. I will keep the positive score as it is.
> >
> > **Weakness (Significance):** I agree that choosing the abstention option should still provide some quantity that enables learning in the theoretical framework -otherwise, the answer to whether having an abstention option improves learning is straightforward. My comment was instead focusing on applying this framework to clinical trials, and in particular on the nature of the "costly prearranged measures" which allow the observation of the output from the *same* stochastic process without having to pay the cost of "classical" testing. I don't see how it translates to real life, so I believe that the practical application of the abstention model is unrealistic (or the model assumptions should be relaxed to match a more realistic setting). My comment is merely a counter-argument to the claimed real-life application (an argument which does not remove any of the theoretical value of this work).
> >
> > **Question 1 (Experiments):** The explanation makes sense, and it is good that a sentence in the main text now refers to this.

---

> > > ### Author Response · Authors · 2023-11-15
> > >
> > > Thank you for providing prompt feedback and affirming the theoretical value of our work.
> > >
> > > In our model, the abstention option exerts no influence on the stochastic observation from the selected arm. Its impact is confined to the instantaneous regret at a specific time step, which we model as either fixed regrets or fixed rewards. We think that in clinical trials, some prearranged measures such as purchasing specialized insurance packages will have a very limited influence on the experimental results, and hence our abstention model is applicable. Our work opens the door to incorporating the abstention option into the multi-armed bandit framework. Delving into more sophisticated and general approaches to model the effect of the abstention option promises to be an intriguing avenue for future research.
> > >
> > > Once again, we express our gratitude for your positive feedback, and we welcome any additional comments or suggestions you may have.

---

### Official Review · Reviewer_4Y9t · 2023-10-25

**Soundness:** 2 fair
**Presentation:** 3 good
**Contribution:** 2 fair
**Rating:** 6
**Confidence:** 4

**Summary:**

This paper considers a multi-armed bandits problem with the choice of abstention. Specifically, after choosing an arm to pull, the algorithm can choose to abstain. In this case, it will observe the random reward, but not receive it. Instead, its reward at this time is a fixed value $c$ (fixed reward setting), or its regret at this time is a fixed value $c$ (fixed regret setting). In both cases, the authors provide algorithms that achieve tight regret upper bounds. They also use experiments to demonstrate the effectiveness of their algorithms.

**Strengths:**

The proofs seem to be correct, and the paper writing is good. The studied problem is interesting.

**Weaknesses:**

My main concern about this paper is that its contribution seems limited. Though the problem setting is interesting, the algorithms (as well as the analysis) seem straightforward, and I do not see novel techniques or insights in it. Besides, no proof sketch is provided in the main text, and there is also no discussion about this paper's theoretical novelty.

I also have concerns about the motivation, e.g., in the example of clinical trials (in the section of the introduction), what is the reason that we do not count the random outcome in reward in the case that we can observe it? In my opinion, the abstention is like that we can pay an extra cost to avoid catastrophic outcomes. This seems more natural than the fixed regret setting or the fixed reward setting.

My next question is why we apply a Thompson Sampling based algorithm but not a UCB-based algorithm? Would a UCB-based algorithm work in this setting? What's its performance in experiments?

**Questions:**

See above "Weaknesses"

---

> ### Author Response · Authors · 2023-11-14
> **Part 1 of Response to Reviewer 4Y9t**
>
> We express our gratitude to the reviewer for providing insightful comments and valuable suggestions on our paper. In response, we have conscientiously attended to all the raised concerns.
>
> ---
>
> **Weakness 1 (Theoretical contribution):**
>
> Regarding the theoretical contribution of our paper, while we submit that the individual ingredients that constitute the algorithms and their analysis are not groundbreaking in the bandit literature, the **adaptation** to  being amenable to incorporating the abstention option (which is a novel setting in the bandit literature) and retaining the asymptotic and minimax optimality of the resultant algorithms is far from straightforward, as acknowledged by both Reviewers n2KC and ahVb.
>
>
> Turning to the significance of our work, our findings offer valuable quantitative insights into the advantages of the abstention option, thereby laying the groundwork for further exploration in other online decision-making problems with such an option. This positive aspect is also recognized by all three other reviewers. Specifically, we have extensively discussed the rationale and insights behind the asymptotic and minimax regret within the main text, offering potential inspiration for addressing analogous challenges in other online learning problems involving abstention.
>
> Due to the density in exposition and comprehensiveness of our results, we struggled with space limitations, which prevent us from incorporating proof sketches in the main text. Nevertheless, we are confident that the proofs presented in the appendix are lucid and easily understandable. For the more intricate fixed-regret setting, we delve into the theoretical challenges associated with designing and analyzing the algorithm in Remark 2, which are decidedly non-trivial and demands meticulous effort.
>
>
> ---
>
> **Weakness 2 (Motivation):**
>
> In the example of clinical trials presented in the introduction, if researchers incorporate costly prearranged measures such as insurance packages, the stochastic treatment outcomes, whether favorable or unfavorable, will not contribute to the optimization objective of the study. Nevertheless, these outcomes remain observable to the researcher and can be utilized to make more effective future experimental designs. If the researcher is unable to observe the outcomes, then conducting medical treatments becomes meaningless.
>
> We do not fully comprehend the meaning of "avoiding catastrophic outcomes", and would appreciate it if the reviewer could provide further comments so that our response is better tailored to the reviewer. In our model, the abstention option is indeed present to mitigate catastrophic outcomes, as the algorithm(s) can utilize the abstention option if it finds that the reward accrued from the chosen arm at a particular time is unacceptably low. Furthermore, the decision to abstain does not influence the stochastic outcome from the selected arm. Its impact is confined to the instantaneous regret at a specific time step, which is modeled as either fixed regrets or fixed rewards. In both the fixed-regret and fixed-reward settings, the instantaneous regret does not depend on the random outcome when the agent chooses to abstain. Exploring a more sophisticated approach to model the cost associated with the abstention option will be an intriguing avenue for future research.
>
>
> ---

---

> ### Author Response · Authors · 2023-11-14
> **Part 2 of Response to Reviewer 4Y9t**
>
> **Weakness 3 (Selection of the base algorithm):**
>
>
> In the fixed-reward setting, our algorithm FRW-ALGwA can leverage any base algorithm that is asymptotically and minimax optimal for canonical multi-armed bandits as its input. In Appendix A, we provide comprehensive definitions of asymptotic and minimax optimality within the canonical model. Eligible candidate base algorithms include both Thompson Sampling-based and UCB-based approaches. Due to space limitations, we defer all experimental results pertaining to the fixed-reward setting to Appendix E. In this appendix, we examine the empirical performances of two particular realizations of our algorithm: FRW-TSwA and FRW-UCBwA. The former uses Less-Exploring Thompson Sampling (Jin et al., 2023) as its base algorithm, while the latter employs KL-UCB++ (Menard \& Garivier, 2017). Notably, while both FRW-TSwA and FRW-UCBwA represent implementations of our general algorithm and share identical theoretical guarantees, FRW-TSwA demonstrates superior empirical performance, as illustrated in Figures 4 and 5.
>
> In the fixed-regret setting, the sampling rule of our algorithm FRG-TSwA is built upon Less-Exploring Thompson Sampling. The fixed-reward setting is inherently more complex than its fixed-regret counterpart because, in the latter setting, we need to dynamically estimate the suboptimality gaps and quantify the regret that results from inaccurately estimating the gaps. These complexities necessitate a more meticulous exploration into the arm sampling dynamics, and preclude us from formulating a generalized strategy. To achieve both forms of optimality, we suspect that the sampling rule should be *anytime* (i.e., not dependent on the time horizon $T$), a criterion not met by current UCB-based approaches. Nevertheless, we do believe it is possible to design a UCB-based algorithm for our abstention model. Finally, regarding the empirical performance, our focus lies on algorithms with solid theoretical guarantees.
>
> > Pierre Menard and Aurelien Garivier. A minimax and asymptotically optimal algorithm for stochastic bandits. In International Conference on Algorithmic Learning Theory, pp. 223–237. PMLR, 2017.
>
>
> ---
>
> Thanks again for your valuable review. Hope these replies resolve your concerns, and any further comments are welcome!

---

> ### Author Response · Authors · 2023-11-19
>
> Dear Reviewer 4Y9t,
>
> We hope this message finds you well. As the rebuttal period draws to a close on November 22nd, just four days away, we would like to reach out to you and inquire if you have any additional suggestions regarding our paper. We would be immensely grateful if you could kindly review our responses to your comments. This would allow us to address any further questions or concerns you may have before the rebuttal period concludes.
>
> We sincerely appreciate the time and effort you have put into reviewing our work and providing valuable feedback. Thank you for your contributions towards improving the quality of our research.
>
> Best regards,
>
> Authors of Submission2358

---

> > ### Comment · Reviewer_4Y9t · 2023-11-20
> > **Reply to rebuttal**
> >
> > About W1. I read all the proofs in appendix. They are indeed easily understandable. However, I still do not understand the novelty of the analysis. All the analysis seems straightforward. Could you please explain this part in detail? What are the challenges and why they are challenges?
> >
> > About W2. I really encourage the authors to try some more complicated settings. Including abstention is an interesting approach, but the current settings seem too simple.
> >
> > About W3. Can you be more specific? Where should we use the property of anytime sampling rule? I guess it is Lemma 5, but I think such results can be easy to get in UCB-based algorithms as well.

---

> > > ### Author Response · Authors · 2023-11-21
> > >
> > > Thank you for providing feedback on our response. We have incorporated some additional follow-ups below.
> > >
> > > ---
> > >
> > > **Weakness 1:**
> > >
> > > Admittedly, some aspects of the analysis follow from previous lines of work. However, this is true for almost every paper, which builds on ideas from other papers. In our case, our analysis hinges on a diverse set of
> > >  techniques from the bandit literature.
> > >
> > > Consider the fixed-regret setting. Since our model is a strict generalization of the canonical multi-armed bandit model (without the abstention option), we construct our sampling rule based on Less-Exploring Thompson Sampling, which is both asymptotically optimal and minimax optimal in the canonical model.  The subsequent challenges for the setting revolve around designing and analyzing the abstention criteria. To preserve the optimality properties of Less-Exploring TS, we must carefully manage the simultaneous control of asymptotic and worst-case regrets resulting from incorrectly opting for the abstention option. During the course of our research, we explored various abstention rules before ultimately selecting the current criteria (Step 5 of Algorithm 1), which is based on the **carefully tailored** lower confidence bounds of the arms ${i\in [K]\setminus \{A_t\}}$. *Indeed, the design process conceals several intricate challenges.* For example, our choice to utilize LCBs rather than certain other natural types of UCBs in the abstention criteria is driven by the necessity to control the magnitude of worst-case regret.
> > >
> > > Besides, given the abstention criteria, the analysis itself is not straightforward. The combination of TS-based sampling rule and LCB-based abstention criteria demands a careful amalgamation of both TS-type and UCB-type analytical techniques. Specifically, it is crucial to establish upper bounds on $\mathbb{E}[N_i^{(1)}(T)]$ for arms $i$ with $\Delta_i <c$, and on $\mathbb{E}[N_i^{(0)}(T)]$ for arms $i$ with $\Delta_i >c$. These bounds need to be derived from both asymptotic and minimax perspectives, adding layers of complexity to the analytical process; see Appendix C.1.
> > >
> > > Thus in summary, the **design** and **analysis** of the various algorithms are non-trivial. In particular, the former is obtained through several painful iterations of trial and error; the finally-presented algorithms are clean precisely because of the careful thoughts that had gone into the design (and writing). We hope that the reviewer agrees with us that clean algorithms, lucid solutions to problems, and clear analyses should be merits in themselves.
> > >
> > >
> > > ---
> > >
> > > **Weakness 2:**
> > >
> > > Our setting is completely novel in the realm of MABs, drawing inspiration from real-world scenarios where decision-makers may wish to deploy prearranged measures, as exemplified in clinical trials. The novel nature of our setting (the first time abstention is explored in the MAB model) not only addresses the current needs but also opens avenues for further exploration and extension by other researchers. For instance, an extension of our framework to linear bandits is anticipated, in which the technical challenges are envisioned to be more substantial, and we leave it for future work (or for other researchers to tackle).
> > >
> > > ---
> > >
> > > **Weakness 3:**
> > >
> > > We are delighted that the reviewer is interested in our speculation that the sampling rule should be anytime. While we wish to underscore that this is purely **speculative** without rigorous evidence, we are eager to delve deeper into the topic. To be specific, the reason we doubt the applicability of not-anytime sampling rules in our setting is that, compared to anytime sampling rules, a larger number of suboptimal arms are likely to be pulled during the early stage. In the early stage, the estimation of the optimal arm lacks sufficient confidence, potentially resulting in incorrectly choosing the abstention option.
> > >
> > > Besides, Lemma 5 is used to establish the asymptotic upper bound in the fixed-regret setting, combined with our abstention criteria. Specifically, it controls the number of times the optimal arm is pulled. We believe it may be possible to design an anytime UCB-based algorithm that satisfies Lemma 5. However, we are currently uncertain about the level of difficulty on this task.
> > >
> > > Finally, we remark that to the best of our knowledge, KL-UCB++ (Menard \& Garivier, 2017) and ADA-UCB (Lattimore, 2018) are both asymptotically and minimax optimal for Gaussian rewards. While we do not favor Thompson Sampling over UCB, we opt for a TS-based approach, since both current UCB-based approaches are not anytime algorithms. Furthermore, we maintain our belief in the feasibility of designing a UCB-based algorithm for our abstention model.

---

> > > > ### Comment · Reviewer_4Y9t · 2023-11-21
> > > > **Thanks for your reply**
> > > >
> > > > For W1. If the algorithm is carefully designed so that it is both asymptotically optimal and minimax optimal (e.g., using LCB of the other arms as well as the empirical mean of the pulled arm when deciding whether to abstain), I suggest the authors to include some detailed explanation about this in the main text. Otherwise it is hard for readers to understand why the task is challenging.
> > > >
> > > > ======================
> > > >
> > > > Thanks for your rebuttal and all the replies. My score is increased to 6.

---

> > > > > ### Author Response · Authors · 2023-11-21
> > > > >
> > > > > Thank you for engaging in discussion with us! In the final version, we will incorporate some of the key points into the main text, provided space allows.

---

### Official Review · Reviewer_ahVb · 2023-10-30

**Soundness:** 4 excellent
**Presentation:** 4 excellent
**Contribution:** 3 good
**Rating:** 8
**Confidence:** 4

**Summary:**

This paper introduces a novel extension of the standard MAB problem to include the option of abstention while pulling an arm: at each arm pull, the agent has the option to abstain from obtaining a stochastic reward and instead accept one of the following 2 options: (a) suffer a fixed regret, or (b) gain a guaranteed reward. For both options, the authors designed and analyzed algorithms whose regrets match the  minimax-optimal and asymptotically optimal lower bounds. Numerical simulations are also provided to corroborate the theoretical results.

**Strengths:**

Originality
----------------------------------------------------------------------------------------------------------------
- The algorithms and analysis are non-trivial extensions of the known methods in the MAB literature

Quality
------------------------------------------------------------------------------------------------------------------
- The analysis seems to be correct

Clarity
-----------------------------------------------------------------------------------------------------------------
- The paper is well-written and easy to follow
- The algorithms and analysis are clearly explained
- Great intuition provided for the behavior of the algorithms and their analysis

Significance
------------------------------------------------------------------------------------------------------------------
- Incorporating the abstention into the standard MAB model lays the groundwork for understanding complex real-world online learning systems (as evident from the motivating example of clinical trials in the paper)

**Weaknesses:**

- Insufficient numerical experiments: While the authors have carefully chosen the arm means $\mu^{\dagger}$ and $\mu^{\ddagger}$ for the experiments, I would like to see the numerical experiments for arbitrary values of arm means and large number of arms.
- "Gaussian" is spelt incorrectly in Lemmas 3 and 4 in Appendix B (independent $\sigma$-sub-"Guassian")
- While proving $(\star)_{i}$ in the asymptotic upper bound for the fixed regret setting (Appendix C), $\beta(b, \mu, K)$ isn't defined until Lemma 5.
- It will be great if the authors can provide a proof sketch of Lemma 5 (instead of referring the readers to Korda et. al. (2013)) in the appendix for the sake of completeness.

**Questions:**

Please see the Weaknesses section

---

> ### Author Response · Authors · 2023-11-14
>
> We thank the reviewer for the positive feedback, valuable comments, and suggestions on our paper. Please find below our response to the  raised concerns.
>
> ---
>
> **Weakness 1 (Insufficient numerical experiments):**
>
> Our paper is primarily positioned as a theoretical work, with numerical experiments serving to validate our theoretical findings. The selection of instances $\mu^{\dagger}$ and $\mu^{\ddagger}$ was not deliberate. In Appendix E.2 of the revised manuscript, we have heeded the reviewer's advice and incorporated additional numerical experiments, using random instances with large numbers of arms. The construction of these random instances follows the method in Jin et al. (2023). Due to time constraints and for the sake of simplicity in presentation, we focus on the fixed-regret setting. It can be seen that the results in Figures 6 and 7 closely resemble those in Figures 2 and 3.
>
>
> ---
>
> **Weaknesses 2 and 3:**
>
> Thank you for bringing the typo and the presentation issue to our attention. In the revision, we have corrected the typo and explicitly clarified that $\beta(b, \mu, K)$ is defined in Lemma 5.
>
>
> ---
>
> **Weakness 4 (Proof Sketch of Lemma 5):**
>
> Thank you for the constructive suggestion. In response, we have incorporated a proof sketch for Lemma 5 in the revised version. It is highlighted in red within the appendix, focusing on the core ideas.

---

> > ### Comment · Reviewer_ahVb · 2023-11-17
> >
> > Thank you for addressing the comments in the Weaknesses section. I am satisfied with the responses and keeping my positive score. It would be great if the authors could also provide the numerical results in the fixed reward setting for arbitrary values of arm means and large number of arms in the revised version of their submission.

---

> > > ### Author Response · Authors · 2023-11-17
> > >
> > > Thank you for the favorable assessment. We are pleased to learn that you found the responses satisfactory. We will certainly include the additional numerical results for the fixed-reward setting in the final version.

---

### Official Review · Reviewer_n2KC · 2023-11-01

**Soundness:** 4 excellent
**Presentation:** 4 excellent
**Contribution:** 3 good
**Rating:** 6
**Confidence:** 3

**Summary:**

This paper introduces a new stochastic multi-armed bandit setting, where instead of just having the option of pulling one of the K arms in each round the learner also has the option to abstain. The authors consider two models: one where abstaining incurs a fixed known reward to the agent and one where abstaining incurs a fixed known regret to the learner. Importantly, in every round the agent specifies both which arm to pull and whether to abstain or not. Even if the agent chooses to abstain, they sill observe a sample of the reward of the arm chosen from the true distribution of the rewards of the arm.

In the fixed regret setting, the authors provide an algorithm that achieves both a minimax optimal and an asymptotically optimal regret bound. Their algorithm builds upon a variant of Thompson sampling by Jin et al. '23, where a natural criterion is added that determines whether the learner should abstain or not. In order to show the optimality of their algorithm the authors extend well-known lower bounds from the stochastic multi-armed bandit literature to the setting they consider.

Similarly, in the fixed reward abstention setting the authors provide a transformation from algorithms that perform well in the traditional multi-armed bandit setting to algorithms that perform well in the setting that the paper considers. In particular, the authors show various instantiations of these transformations that obtain optimal minimax and asymptotic regret bound. In order to establish the optimality of their upper bounds, the authors prove matching lower bounds.

**Strengths:**

- Learning with the option of abstention is an interesting line of work that the learning theory community is excited about. To the best of my knowledge, this is the first paper that considers this problem in the context of multi-armed bandits.

- The algorithms that are provided are clean and elegant.

- The result are interesting, and even though the techniques are a natural adaptation of known results to the abstention setting, the analysis is not entirely straightforward.

- The authors present their results in a clear manner, without overselling their contributions.

**Weaknesses:**

- I think the main weakness has to do with the abstention model that the authors consider. While both having an option that gives a fixed reward or incurs a fixed regret to the learner seem natural to me, being able to observe the sample from the arm that was chosen when the agent chooses to abstain feels a bit too strong. I think the authors need to elaborate a bit on this assumption they have made.


Not weaknesses, but I'm writing some small issues that could be improved.

- It would be nice if the authors could give longer sketches of the proofs in the main body, even though I understand that there isn't enough space. One suggestion would be to move all the experiments to the appendix, since this is mostly a theoretical work and the algorithms are straightforward to implement I don't see what value the experiments add to the paper.

- Since the fixed reward setting is easier to analyze it might make sense to change the order of the presentation between sections 3, 4.

**Questions:**

1) Please see the main weakness.

2) If we assume that when the learner abstains they don't observe the sample from the arm they chose, what is the regret bound they get?

3) Is there a way to state the regret bound of algorithm 2 as a function of the regret of the base algorithm? Since this is a black-box transformation I would expect to see this type of result.

---

> ### Author Response · Authors · 2023-11-14
>
> We thank the reviewer for the insightful comments and suggestions on our paper. We have carefully addressed your concerns in the following.
>
>
> ---
>
> **Weakness (The abstention model):**
>
>
> We consider the assumption that the agent can observe the sample regardless of the abstention decision to be rather natural to our problem setting of bandits with abstentions and in particular to the clinical trials example presented in the introduction. In the example of clinical trials, if researchers implement costly prearranged measures, the treatment outcomes will not contribute to the optimization objective of the study. However, the researcher can still observe these outcomes, thereby leading to more efficient future experimental designs. Furthermore, conducting medical treatments becomes meaningless if the researcher is unable to observe the outcomes. In line with this model assumption, our model well aligns with that of Neu \& Zhivotovskiy (2020), which explored the role of abstention in the context of online prediction with expert advice.
>
>
> Regarding the learnability of the abstention bandit model, if the agent cannot observe the sample from the chosen arm when opting for abstention, then it is equivalent to skipping the time step, rendering the model trivial. In other words, the learner gains no information from the time step but incurs an instantaneous regret.  Finally, we would like to remark that the above statement holds true only for the stochastic bandit model. The dynamics change in non-stationary or adversarial bandits, providing intriguing avenues for future research.
>
> > Gergely Neu and Nikita Zhivotovskiy. Fast rates for online prediction with abstention. In Conference on Learning Theory, pp. 3030–3048. PMLR, 2020.
>
>
> ---
>
> **Small Issue 1 (Longer proof sketches):**
>
> We sincerely thank you for your valuable suggestion and understanding. Due to the density of our results, we, as with many other authors, struggled with space limitations, which prohibits us from incorporating comprehensive proof sketches in the main text. Our paper is positioned as a theoretical work, with numerical experiments serving to validate our theoretical findings. Opinions on the inclusion of experiments in the main text vary, and one reviewer suggests more numerical experiments. In light of this, we have chosen not to relocate all experiments to the appendix. According to the ICLR guidelines, the page limit for the main text is strict, and no additional pages can be added in the final version. Consequently, we are unable to include longer proof sketches in the main body. However, we are confident that the proofs presented in the appendix are lucid and easily understandable.
>
>
>
>
> ---
>
> **Small Issue 2 (Order of the presentation):**
>
> We express our gratitude to the reviewer for providing the constructive suggestion. While preparing the manuscript, we ensured clarity by presenting comprehensive explanations in the first setting and avoiding unnecessary repetitions in the second setting. In particular, in the corresponding proofs, we typically streamlined the common components. Although we appreciate the reviewer's kind suggestion, when preparing the initial manuscript, we have thought through whether to present the fixed-regret or fixed-reward settings first, aiming to ensure that the overall paper is as concise as possible.
>
>
> ---
>
> **Question 2 (Regret bound for the new model):**
>
> This point pertains to the point of weakness above. In the context of stochastic multi-armed bandits, if the agent cannot observe the sample from the chosen arm when opting for abstention, then no information is gained during that time step, rendering it equivalent to skipping the time step altogether. Consequently, the abstention option becomes a means to shorten the time horizon with costs. Regarding the cumulative regret bound during time steps when the agent chooses not to abstain, it is the same as that of the canonical bandit model. Hence, there is nothing to analyze in this setting.
>
> ---
>
> **Question 3 (Regret bound of Algorithm 2):**
>
> Thank you for the question. Since the base algorithm is asymptotically optimal (see Appendix A for relevant definitions), our model that possesses the abstention option and leverages the base algorithm, *inherits the asymptotic optimality of the base algorithm*. Hence, our regret bound is indeed a function of that of the base algorithm but in a rather unusual way; both of them are asymptotically optimal.
> Concerning the minimax upper bound, we indeed possess a result of that type, specifically expressed as a function of the minimax upper bound of the base algorithm. For further details, please refer to the concluding lines of the proof of Theorem 3 in Appendix D.1.

---

> ### Author Response · Authors · 2023-11-19
>
> Dear Reviewer n2KC,
>
> We hope this message finds you well. As the rebuttal period draws to a close on November 22nd, just four days away, we would like to reach out to you and inquire if you have any additional suggestions regarding our paper. We would be immensely grateful if you could kindly review our responses to your comments. This would allow us to address any further questions or concerns you may have before the rebuttal period concludes.
>
> We sincerely appreciate the time and effort you have put into reviewing our work and providing valuable feedback. Thank you for your contributions towards improving the quality of our research.
>
> Best regards,
>
> Authors of Submission2358

---

> > ### Comment · Reviewer_n2KC · 2023-11-22
> >
> > I would like to thank the authors for carefully addressing my comments. I do not have any further questions and I remain positive about the paper.

---

### Meta-Review · Area_Chair_DPDv · 2023-12-12

**Metareview:**

This paper considers the standard multi-armed bandit problem with the additional tweak that the agent can "refuse" the reward proposed by an arm, obtaining instead a fixed reward of regret.

This is a cute setting, that is reminiscent of the traditional one-arm bandit problem.

The reviewers were rather positive about this paper, as it is certainly nicely written, and I was curious about the setting hence I read it myself. Based on it, I acknowledge that it is well written, but unfortunately, I found it quite incremental. I am not saying that the techniques and results are necessarily straightforward, but rather that they are not surprising and the approach considered would be the first someone familiar with bandits would consider.

We had some discussions about these facts, and my opinion was confirmed by expert colleagues. So, unfortunately, I do not think that this paper actually reaches the ICLR bar.

**Justification For Why Not Higher Score:**

The contributions are interesting, but too incremental.

**Justification For Why Not Lower Score:**

N/A

---

### Decision · Program_Chairs · 2024-01-16

Reject